## METHOD

# PhyloMed: a phylogeny-based test of mediation effect in microbiome

Qilin Hong[1], Guanhua Chen[1] and Zheng-Zheng Tang[1*]

*Correspondence:
tang@biostat.wisc.edu

[1] Department of Biostatistics and Medical Informatics, University of Wisconsin-Madison, Madison, WI 53715, USA

**Abstract**

Microbiome data from sequencing experiments contain the relative abundance of a large number of microbial taxa with their evolutionary relationships represented by a phylogenetic tree. The compositional and high-dimensional nature of the microbiome mediator challenges the validity of standard mediation analyses. We propose a phylogeny-based mediation analysis method called PhyloMed to address this challenge. Unlike existing methods that directly identify individual mediating taxa, PhyloMed discovers mediation signals by analyzing subcompositions defined on the phylogenic tree. PhyloMed produces well-calibrated mediation test *p*-values and yields substantially higher discovery power than existing methods.

**Keywords:** Composite null hypothesis, Mediation analysis, Microbiome, Phylogenetic tree

## Background

Recent studies suggest that the human microbiome is emerging as a crucial mediator between treatments (or exposures) and health outcomes. For example, gut microbes are found to greatly influence the potency of immunotherapy and some chemotherapies with immunostimulatory functions in the treatments for cancer [1]. Mediation analysis provides the statistical framework to investigate if a treatment or exposure affects an outcome through a mediator. The traditional meditation analysis studies the mediation effect of a single mediator [2, 3]. More recent developments have extended that to multiple and even high-dimensional mediators [4–8]. A causal interpretation of the mediation effect can be established based on the potential outcomes framework for causal inference [9–11].

Microbiome data from sequencing experiments contain the relative abundance of a large number of different microbes. The compositional and high-dimensional nature of the microbiome mediator poses a significant challenge to traditional mediation analyses. A typical amplicon sequencing study generates hundreds of microbial taxonomic units, such as operational taxonomic units (OTUs) [12] and amplicon sequence variants (ASVs)

[13]. The abundance of each taxon is essentially measured in a fraction. We just know what percent of each taxon made up the total but cannot quantify the actual microbial load (i.e., absolute abundance). In differential abundance analysis, where microbial compositions are compared between samples, methods that do not account for compositionality can result in high false discovery rates [14, 15]. Several composition-aware methods compare the ratio of taxa as the bias introduced by unknown microbial load cancels out after taking the ratios between taxa [14, 16].

MedTest [17] and MODIMA [18] are two distance-based methods for testing the mediation effect of the entire microbial community. These methods summarize the microbial composition into between-sample distance matrices and construct the mediation test statistic using the distance matrices. Therefore, they are not designed to identify mediating taxa. The distance-based tests can achieve good power when many microbial taxa in the community mediate the treatment effect on the outcome. However, their power is limited when the number of mediating taxa is small. LDM-med [19] tests the mediation effect at individual taxa using the relative abundance and combine test statistics across taxa to produce a global test. False discovery rate (FDR)-controlling procedures are applied to identify mediating taxa.

Other mediation analysis methods for microbiome data assume sparse mediation effects and estimate the effects via regularization [20–23]. They aim to select mediating taxa and some [20, 21] provide global tests of the overall mediation effect at the community level. These methods apply different treatments to handle compositional data. CMM [20] uses the composition operators to define the mediation model with the parameters interpreted under the additive log-ratio transformation; microHIMA [22, 23] uses the isometric log-ratio to transform the relative abundance to variables in the Euclidean space; SparseMCMM [20] uses the Dirichlet regression to model microbial compositions. These methods also employ different mediation tests. CMM uses a Sobel-type test [24]; microHIMA uses a joint-significance-type test [25]; SparseMCMM includes two tests: one uses the overall mediation effect estimate as the test statistic and the other uses the sum of squares of the component-wise mediation effect estimates as the test statistic [7], both of which have conservative control of type I error reported in the original paper [21].

In this article, we develop a phylogeny-based mediation analysis method (PhyloMed) for the high-dimensional mediator of microbial composition. Microbial taxa are evolutionarily related and their relationship is represented by a phylogenetic tree. Incorporating phylogenetic information has been shown to improve the performance of various statistical analyses [26–28]. PhyloMed models the microbiome mediation effect through a cascade of independent local mediation models of subcompositions on the internal nodes of the phylogenetic tree. Figure 1 shows an example of a simple rooted binary phylogenetic tree with 11 microbial taxa at leaf nodes and 10 internal nodes, representing the common ancestors of those taxa. The subcomposition on an internal node consists of the relative abundance aggregated at its two child nodes. For example, the highlighted *j*th internal node in Fig. 1 has taxon 5 being its left child node and the most recent common ancestor of taxa 6 and 7 being its right child node. Therefore, the subcomposition defined on that internal node consists of $(M_j, 1 - M_j)$, where $M_j = \frac{v_5}{v_5+v_6+v_7}$, and $v_5$, $v_6$, and $v_7$ are relative abundances of taxa 5, 6, and 7. As depicted in Fig. 1 and

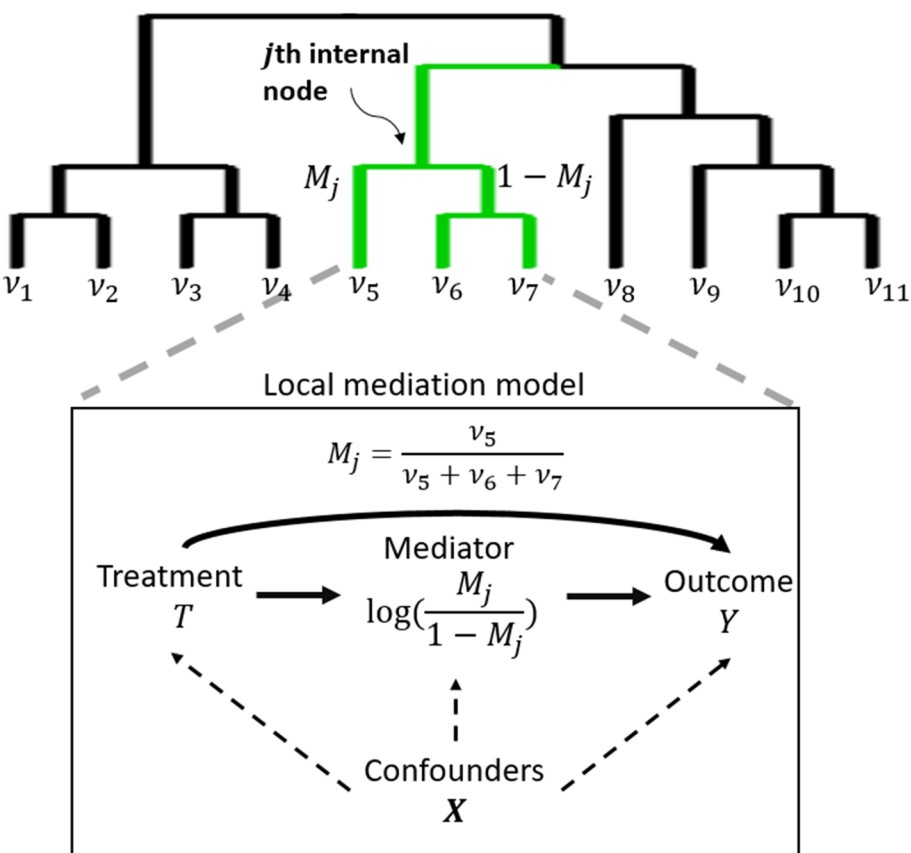

**Fig. 1** Example phylogenetic tree and causal path diagram of a local mediation model. The $v_1$-$v_{11}$ are relative abundances of the taxa on leave nodes. PhyloMed considers subcompositions at every internal node of the phylogenetic tree and tests the mediation effect of the subcompositions

detailed in the "Methods" section, on each internal node of the phylogenetic tree, we propose to construct a local subcomposition mediation model using the log ratio of the abundance on the left and right child nodes (i.e., $\log(\frac{M_j}{1-M_j})$) as the mediator variable.

We develop a testing procedure to ensure the PhyloMed local mediation test *p*-values are asymptotically mutually independent and uniformly distributed under the global null hypothesis that no mediation effect in any of the internal nodes (Methods). We apply a FDR-controlling procedure to the local mediation *p*-values and identify mediating internal nodes. The descendants of an identified node represent a group of evolutionarily close taxa that potentially plays a mediating role. Furthermore, we can combine local mediation *p*-values over internal nodes to test the global mediation effect of the entire microbial community. In this article, we employed the Benjamini-Hochberg (BH) procedure [29] in identifying mediating internal nodes and the harmonic mean *p*-value (HMP) combination method [30] in the global mediation test.

PhyloMed models mediation effects in many subcompositions on the phylogenetic tree rather than in a full composition with many taxa at the low taxonomic rank. There are several advantages of working with subcomposition mediation models. First, it avoids the difficulty of jointly modeling a large number of microbial taxa. Second, the descendants of each internal node share a certain degree of evolutionary affinity and

tend to have similar biological functions. Therefore, mediating taxa are likely to cluster on the tree, and PhyloMed can potentially enrich mediation signal and improve the power of mediation analysis. Third, as we explain next, the large number of independent subcomposition mediation models enables us to obtain well-calibrated mediation test *p*-values and boost discovery power.

In each local model of PhyloMed, we test the mediation effect of a subcomposition in the treatment-to-outcome pathway. The mediation effect is commonly expressed as a product of two parameters, the treatment-mediator association ($\alpha$) and the mediator-outcome association ($\beta$) conditional on the treatment. The null hypothesis of no mediation effect is composite: either one of those associations is zero or both are zeros ($H_{00} : \alpha = \beta = 0$ or $H_{10} : \alpha \neq 0, \beta = 0$ or $H_{01} : \alpha = 0, \beta \neq 0$). Traditional tests, such as the Sobel's test [24] and the joint significance test, are overly conservative and yield low power because they ignore the composite nature of the null hypothesis [25, 31]. To address this problem, we estimate the proportions of different nulls ($H_{00}, H_{10}, H_{01}$) among all the local mediation models and obtain the subcomposition mediation test *p*-value using a mixture distribution with three components, each corresponding to one type of null hypothesis (Methods). A large number of independent subcomposition mediation tests enables us to accurately estimate the proportions of different nulls. Therefore, the high-dimensionality of microbiome data becomes a blessing instead of a curse in PhyloMed.

One needs to be cautious about the separation of the treatment-mediator associational element and mediator-outcome associational element when interpreting the mediation signals identified from the microbiome data. The mediation effect at the ancestor taxon can be formed by aggregating lower-level taxa associated with only the treatment and lower-level taxa associated with only the outcome (i.e., the separation of two elements demonstrated in the last column of Additional file 1: Fig. S1). Therefore, if we want to extend the interpretation of mediation signal (discovered by PhyloMed or any other methods) from the upper level to the lower level, we need to assume that at least one descendant at the lower level has inherent the mediation effect from the mediating ancestor (i.e., no complete separation of the two elements demonstrated in the third column of Additional file 1: Fig. S1). In PhyloMed, we test the mediation effects at internal nodes but not at the leaf-level taxa. We show in the real data application a heuristic approach to investigate the mediation effect at few leaf-level taxa under the PhyloMed-identified internal nodes.

The separation of the two elements can occur even when we analyze taxa at one taxonomic level because of the compositional nature of the microbiome mediator. The relative abundances of all taxa are linked because of the unit-sum constraint of the proportions across taxa: changing the absolute abundance of one taxon would shift the relative abundance of all other taxa. Consequently, if a taxon is identified as a mediator using its relative abundance in the mediation model, the two elements contributing to the mediation signal may come from entirely different sets of taxa. To address compositionality, many methods assume the mediation signal is sparse and apply different transformations to the relative abundance. The mediation effects are defined and interpreted in the context of a particular treatment of the compositional data. We need the assumption of no complete separation of the two elements in absolute abundance if we want

to extend the interpretation of the identified mediation signal to the absolute abundance level. This is probably a reason why many mediation methods for microbiome data focus more on hypothesis testing than estimation. The test results provide a scan of high-dimensional microbial composition and generate candidates for downstream validation studies and mechanistic experiments.

## Results

### Simulation studies

We performed extensive simulation studies to evaluate the performance of PhyloMed under different settings. The simulation strategy is detailed in Methods. In short, we used the top 100 most abundant OTUs and the phylogenetic tree from a real gut microbiome study [32] as a basis. We considered the sample size of 50 or 200, a binary treatment variable that defines a treatment and a control group, and two types of outcome variables (continuous and binary). Association and mediation signals were added at the OTUs. We let $\mathcal{S}_\alpha$ and $\mathcal{S}_\beta$ denote the set of treatment-associated OTUs and outcome-associated OTUs, respectively. To perturb the abundance of a treatment-associated OTU, we added random counts to the subjects in the treatment or control group. To generate values of outcome, we used the log-contrast model [33] with the outcome-associated OTUs as covariates. Under the null hypothesis of no mediation effect, the $\mathcal{S}_\alpha$ and $\mathcal{S}_\beta$ do not overlap, and we considered different mixtures of nulls by adjusting the OTUs in the two sets. Under the alternative, both sets contain the mediating OTUs. We varied the number of mediating OTUs and made them clustered on the tree or randomly scattered.

### *PhyloMed controls type I error and improves power in the global mediation test of the microbial community*

PhyloMed local mediation test statistic is constructed as the maximum of the treatment-mediator association test $p$-value and the mediator-outcome association test $p$-value (Methods). We obtain these $p$-values via asymptotic approximation or permutation (Methods) and refer to the corresponding PhyloMed result as PhyloMed.A and PhyloMed.P. We compared the performance of PhyloMed global mediation test with the global tests in MedTest [17], MODIMA [18], LDM-med [19], and CMM [20]. For the distance-based tests, we considered Aitchison, Bray-Curtis, Jaccard, weighted, and unweighted UniFrac distances. We reported the omnibus MedTest test over the five distances. As MODIMA does not provide an omnibus test, we reported the Bonferroni-corrected minimal $p$-values among the five distances. The CMM method is for the continuous outcome. We found that CMM often fails to converge when the sample size is smaller than the number of taxa. Therefore, the CMM result is only reported in the setting of continuous outcome and large sample size ($n = 200$).

The control of the type I error is demonstrated by the quantile-quantile plots of $p$-values from different global tests under various null hypothesis settings (Fig. 2 and Additional file 1: Fig. S2). The PhyloMed global test yields $p$-values that are mostly aligned with the diagonal line suggesting that the test produces uniformly distributed $p$-values under the null hypothesis of no mediation effect. The empirical type I error of PhyloMed is much closer to the significance level than the other methods in

**Table 1** Empirical type I error of different global mediation tests (significance level = 0.05). The $|\mathcal{S}_\alpha|$ and $|\mathcal{S}_\beta|$ denote the number of treatment-associated OTUs and outcome-associated OTUs, respectively. Different combinations of ($|\mathcal{S}_\alpha|, |\mathcal{S}_\beta|$) represent different mixtures of mediation nulls $H_{00}$, $H_{10}$ and $H_{01}$

| $n$ | $|\mathcal{S}_\alpha|$ | $|\mathcal{S}_\beta|$ | PhyloMed.A | PhyloMed.P | MedTest | MODIMA | LDM-med | CMM |
|---|---|---|---|---|---|---|---|---|
| **Continuous outcome** | | | | | | | | |
| 50 | 0 | 0 | 0.020 | 0.028 | 0.002 | 0.002 | 0.005 | - |
| 50 | 3 | 0 | 0.021 | 0.028 | 0.013 | 0.007 | 0.006 | - |
| 50 | 6 | 0 | 0.036 | 0.042 | 0.025 | 0.009 | 0.009 | - |
| 50 | 0 | 3 | 0.021 | 0.025 | 0.003 | 0.002 | 0.009 | - |
| 50 | 0 | 6 | 0.024 | 0.028 | 0.011 | 0.003 | 0.016 | - |
| 200 | 0 | 0 | 0.022 | 0.024 | 0.001 | 0.000 | 0.002 | 0.308 |
| 200 | 3 | 0 | 0.030 | 0.032 | 0.022 | 0.008 | 0.013 | 0.495 |
| 200 | 6 | 0 | 0.035 | 0.036 | 0.032 | 0.023 | 0.015 | 0.532 |
| 200 | 0 | 3 | 0.032 | 0.034 | 0.015 | 0.006 | 0.017 | 0.210 |
| 200 | 0 | 6 | 0.040 | 0.043 | 0.028 | 0.015 | 0.021 | 0.154 |
| **Binary outcome** | | | | | | | | |
| 50 | 0 | 0 | 0.017 | 0.020 | 0.000 | 0.000 | 0.004 | - |
| 50 | 3 | 0 | 0.028 | 0.033 | 0.011 | 0.002 | 0.010 | - |
| 50 | 6 | 0 | 0.029 | 0.037 | 0.023 | 0.011 | 0.012 | - |
| 50 | 0 | 3 | 0.021 | 0.031 | 0.002 | 0.000 | 0.006 | - |
| 50 | 0 | 6 | 0.018 | 0.029 | 0.004 | 0.002 | 0.007 | - |
| 200 | 0 | 0 | 0.018 | 0.021 | 0.002 | 0.000 | 0.003 | - |
| 200 | 3 | 0 | 0.043 | 0.044 | 0.025 | 0.011 | 0.009 | - |
| 200 | 6 | 0 | 0.040 | 0.040 | 0.030 | 0.016 | 0.018 | - |
| 200 | 0 | 3 | 0.024 | 0.028 | 0.008 | 0.002 | 0.005 | - |
| 200 | 0 | 6 | 0.038 | 0.045 | 0.018 | 0.007 | 0.011 | - |

all scenarios (Table 1). The permutation version of PhyloMed (PhyloMed.P) controls type I error slightly better than the asymptotic version (PhyloMed.A) when the sample size is small ($n = 50$). MedTest, MODIMA, and LDM-med tests are generally conservative. When all taxa are under $H_{00}$, their conservativeness is worse than in other scenarios where some taxa are under $H_{10}$ or $H_{01}$. In contrast, the performance of the PhyloMed global test is less affected by the mixture proportions of different nulls. The CMM test is too liberal and results in many false positives in our simulation study.

Figure 3 displays the power results under the setting where the mediating OTUs are clustered on the tree. PhyloMed is more powerful than MedTest, MODIMA, and LDM-med. PhyloMed is also more powerful than CMM, even though CMM's power is overestimated due to the inflation of its type I error. The power gain is because PhyloMed tests subcompositions on the ancestor nodes of mediating OTUs, at which mediation signals were condensed. Moreover, PhyloMed employs the mixture distribution in testing the subcomposition mediation effect, which is more efficient than traditional mediation tests. Additional file 1: Fig. S3 displays the power results when including more OTUs in the simulation. In this case, all methods become less powerful, but the power of PhyloMed remains the highest. Additional file 1: Fig. S4 displays the power results when the mediating OTUs are randomly scattered. PhyloMed still has a substantial power gain over other methods in most scenarios, especially when the mediation signal is sparse.

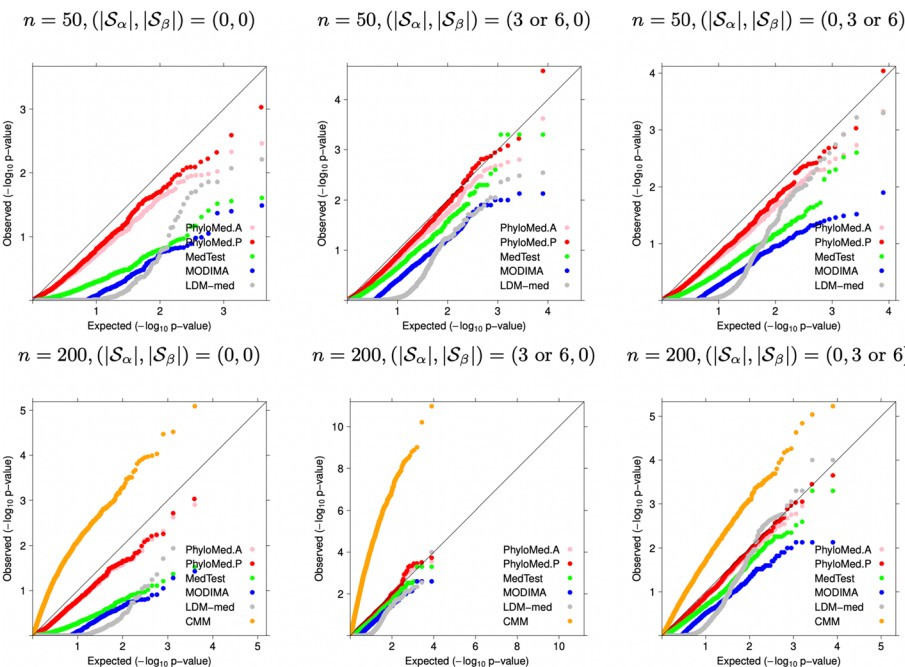

**Fig. 2** Quantile-quantile plots of *p*-values from different global mediation tests in the simulation study for the continuous outcome. The observed *p*-values were compared to the expected quantiles generated by the uniform null distribution. The $|\mathcal{S}_\alpha|$ and $|\mathcal{S}_\beta|$ denote the number of treatment-associated OTUs and outcome-associated OTUs, respectively. Different combinations of $(|\mathcal{S}_\alpha|, |\mathcal{S}_\beta|)$ represent different mixtures of mediation nulls $H_{00}$, $H_{10}$, and $H_{01}$

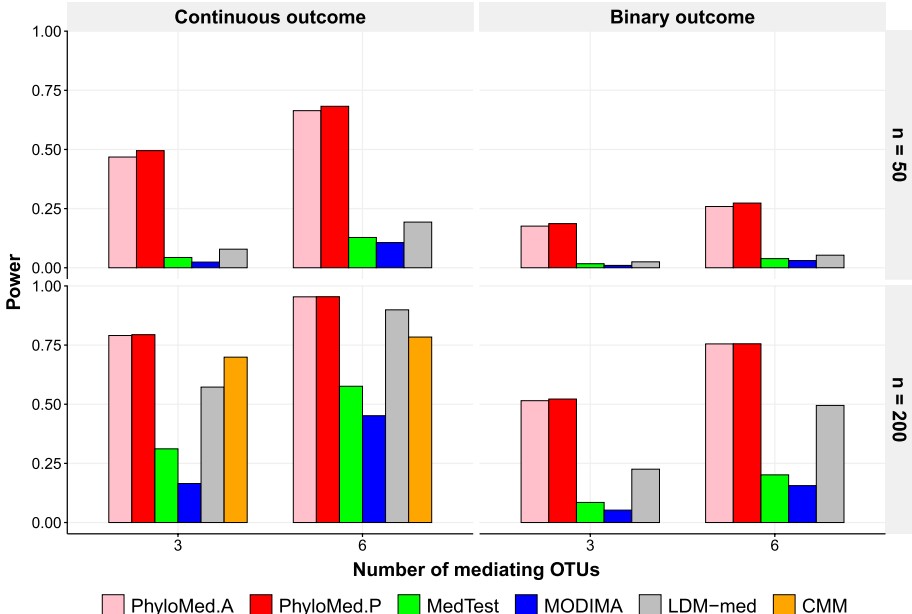

**Fig. 3** Power comparison of different global mediation tests when mediating OTUs are clustered on the tree

### PhyloMed powerfully detects microbial subcompositions with mediation effects

We also evaluated the empirical FDR and power in identifying mediating nodes on the tree at the target FDR of 0.05. All the ancestor nodes of the mediating OTUs are

mediating nodes. To evaluate power, we focused on the discovery rate of identifying the most recent common ancestor of the mediating OTUs because the signal on the upper-level ancestor nodes was diluted by the OTUs with no mediation effects and is difficult to detect by any method. Besides the mixture-distribution-based test implemented in PhyloMed, we employed the Sobel's test and the joint significance test to obtain the *p*-value for testing the subcomposition mediation effect in each local model.

In detecting mediating nodes, the empirical FDR is controlled for all local mediation tests (Additional file 1: Table S1). We also evaluate the empirical FDR with varying mediation signal strength and density. The empirical FDR is controlled for all signal strength and density levels, even though BH controls the FDR at a more stringent level for stronger and denser signals (Additional file 1: Fig. S5). Additional file 1: Table S2 shows the discovery rate of the most recent common ancestor of the mediating OTUs. PhyloMed has much higher power than the Sobel's test and the joint significance test in identifying the mediating node.

### Application of PhyloMed to study the mediation of mouse cecal microbiome in the relationship between antibiotics treatment and body fat

We demonstrate the utility of PhyloMed by analyzing a cecal microbiome dataset from a randomized mouse experiment [34]. The study randomly assigned mice to four types of antibiotics and a control group and evaluated the group difference in body fat percentage and cecal microbial composition. The mice in the antibiotics group have a higher average body fat percentage than those in the control group (Additional file 1: Fig. S6). We are interested in investigating if the cecal microbiome mediates the effect of antibiotics on body fat. After removing samples with low sequencing depth, we had 48 samples (38 in antibiotics and 10 in controls). Due to the small sample size of this study, we included in the analysis the top 100 most abundant OTUs with at least 20% non-zero observations as potential mediators. These taxa make up 80% of the cecal microbial composition.

The PhyloMed global mediation test (PhyloMed.P) gives a *p*-value of 0.085. MedTest, MODIMA, and LDM-med global tests give *p*-values of 0.24, 1, and 0.42. PhyloMed also pinpoints two internal nodes with significant mediation effects on the phylogenetic tree while controlling FDR at 0.1. When we employed the Sobel's test and the joint significance test in local models, the subcomposition mediation test *p*-values showed a pattern of deflation, suggesting low power (Fig. 4b), and no mediating nodes were identified. We also applied LDM-med and microHIMA to identify mediating OTUs and no OTU was selected at FDR = 0.1.

It is instructive to examine the internal node with the smallest PhyloMed local mediation *p*-value. This node has an OTU as its left child node and the most recent common ancestor of the other five OTUs as its right child node (Fig. 4a). The proportion of the abundance at the left child node to the right child node is significantly associated with the antibiotics treatment and the body fat (Fig. 4c). The OTU descendants of the node are a group of evolutionarily close taxa that likely share similar biological functions and jointly contribute to the mechanism of antibiotics' effect on body fat change.

We used a heuristic approach to explore what OTU descendants may contribute to the significant subcomposition mediation on that internal node. In particular, we constructed multiple subcompositions by pairing the OTU at the left child node with

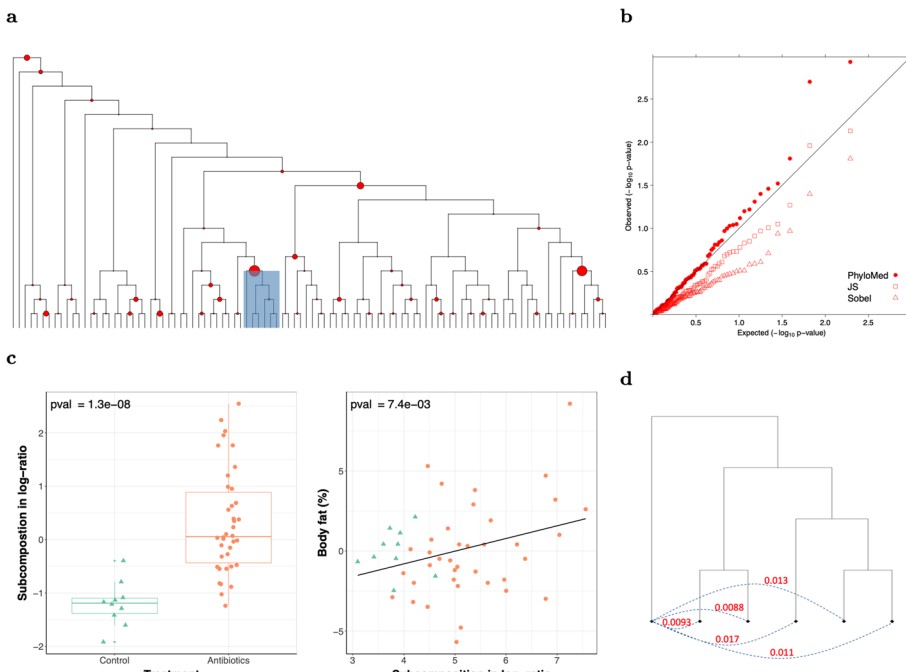

**Fig. 4** Analysis of mouse cecal microbiome data. **a** Phylogenetic tree with the size of the circle at each internal node proportional to − log 10 (PhyloMed local mediation test *p*-value). The identified internal node with the minimum subcomposition mediation test *p*-value and the subtree under the node are highlighted in a blue rectangle. **b** Quantile-quantile plots of different subcomposition mediation test *p*-values across internal nodes. PhyloMed: proposed mixture-distribution-based test; JS: joint significance test; Sobel: Sobel's test. **c** Associations of the subcomposition (in log-ratio) at the highlighted node in **a** with the treatment and the body fat percentage after adjusting for the treatment effect. **d** The highlighted subtree in **a**. Pair the OTU under the left child node to each of the five OTUs under the right child node. For the subcomposition of each pair, examine the treatment-subcomposition and subcomposition-outcome associations and report the maximum *p*-value of the two associations

individual OTU descendants of the right child node and examined the treatment-subcomposition and subcomposition-outcome associations. All the (left OTU and right OTU) pairs exhibit evidence of mediation effect (the maximum *p*-values of the treatment-subcomposition and subcomposition-outcome associations for all the pairs are less than 0.05, see Fig. 4d). This result demonstrates that testing the mediation effect on the internal node of the phylogenetic tree can enrich mediation signals clustered on the tree and boost the power of mediation analysis.

### Application of PhyloMed to study the mediation of human gut microbiome in the relationship between fat intake and body mass index

We also applied PhyloMed to a human gut microbiome dataset from an observational study with a larger sample size [35]. The study sequenced gut microbiome from fecal samples of 96 healthy subjects and collected their diet and health information. The subjects with higher fat intake generally have higher body mass index (BMI) values (Additional file 1: Fig. S7). We used PhyloMed to investigate if the gut microbiome mediates the effect of fat intake on BMI increase. We used 395 OTUs that have at least 20% nonzero observations. These OTUs make up 79% of the microbial community composition. Total calorie intake was adjusted as a potential confounder in the mediation analysis.

The PhyloMed global mediation test (PhyloMed.P) gives a *p*-value of 0.047, indicating the gut microbial community plays a mediating role in the effect of fat intake on BMI increment. MedTest, MODIMA, and LDM-med global tests give *p*-values of 0.68, 0.27, and 1. PhyloMed identifies one internal node with significant mediation effects while controlling FDR at 0.1 (Additional file 1: Fig. S8). The left and right child nodes of the identified internal node are leaf nodes with two OTUs in the *Lachnospiraceae* family. It has been reported that members of the *Lachnospiraceae* family are associated with high-fat diets and diet-induced obesity [36, 37]. Subjects with low fat intake and BMI show dramatically different proportions of the two OTUs compared to those with higher fat intake and BMI (Fig. 5). The log-ratio abundance between the two OTUs is significantly associated with fat intake (*p*-value=$5.9 \times 10^{-3}$) and BMI (*p*-value=$2.2 \times 10^{-3}$). No mediating OTU was identified by LDM-med and microHIMA at FDR = 0.1. The analysis demonstrates that PhyloMed can detect the sparse mediation signal among a large number of taxa, which is often missed by other methods.

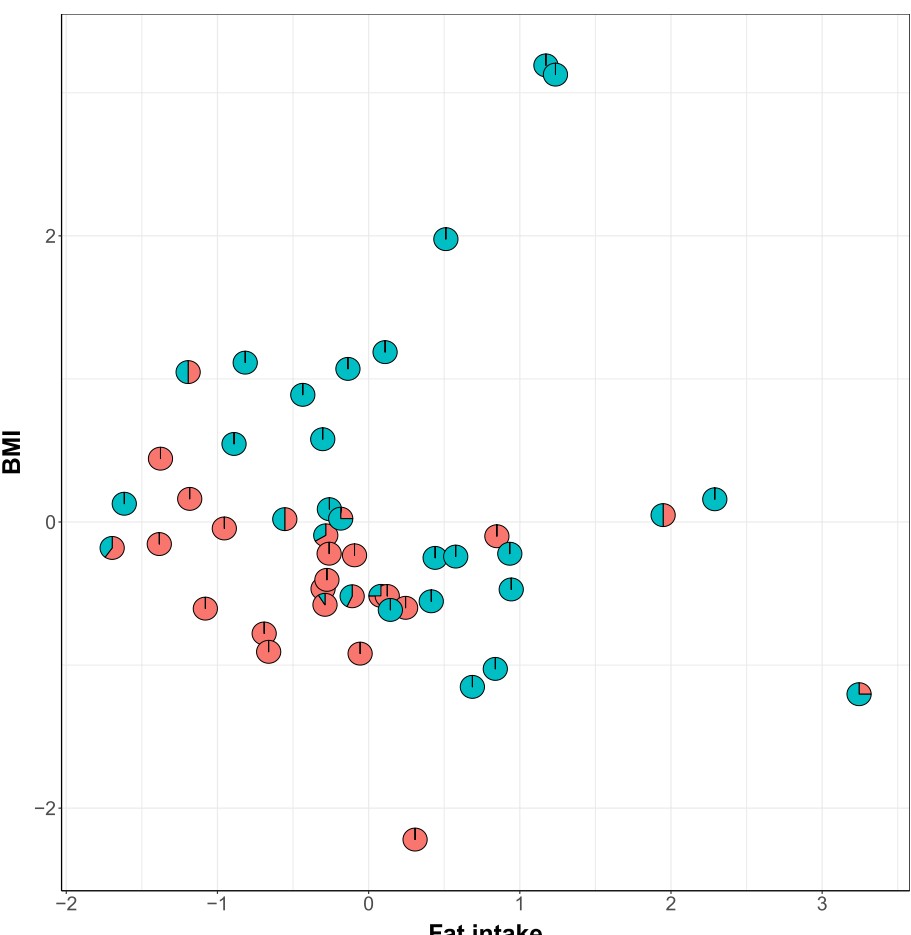

**Fig. 5** Scatter pie plot for the subcomposition at the identified internal node in human gut microbiome data analysis. Each point is a pie chart showing a subject's subcomposition of the two OTUs under the identified internal node. The fat intake value (after adjusting for total calorie intake) is on the *x*-axis and the BMI value (after adjusting for fat intake and total calorie intake) is on the *y*-axis

## Discussion

This article has introduced a new framework PhyloMed for testing the mediation effect in high-dimensional microbial composition. PhyloMed leverages the hierarchical phylogeny relationship among different microbial taxa to decompose the complex mediation model on the full microbial composition into multiple simple independent local mediation models on subcompositions. This tree-guided approach effectively enriches mediation signals that tend to cluster on the phylogenetic tree and boosts the power of the test for weak mediation effects among taxa. Moreover, PhyloMed accounts for the compositional nature of the relative abundance data and the composite mediation null hypothesis, resulting in well-calibrated $p$-values for testing local subcomposition mediation effects. Our simulation studies have shown that PhyloMed properly controls the type I error and has substantially higher power than existing methods. In addition, PhyloMed can pinpoint internal nodes at which the subcompositions have significant mediation effects. The cluster of descendant taxa under the identified node could serve as the top candidate mediating taxa for future biological validation.

Many $p$-value combination methods and FDR-controlling procedures can be applied to the local mediation $p$-values to test global mediation and identify mediating internal nodes. In this article, we adopt the HMP $p$-value combination method that is powerful to detect the sparse signal. If the mediation signal is dense (i.e., many taxa in the microbial community have mediation effects), alternative methods such as Fisher's $p$-value combination method could be more powerful. In addition, advanced FDR methods [38–40] may yield better performance in detecting mediating subcompositions on the internal nodes if mediation hypotheses defined on the nearby nodes are more likely to be jointly true or false. Evaluating these different methods under the PhyloMed framework would be an interesting area for future research.

PhyloMed uses the observed proportion data and does not model the sampling variability of the read counts. A Binomial sampling layer could be added on top of each subcomposition model. However, the benefit of this practice may not outweigh the drawbacks. Although the mediation model can be changed without disrupting the PhyloMed testing procedure, it is not clear if the parameter we test still reflects the size of the causal mediation effect. Even if this is true, modeling the count data would increase the computation burden and may not readily improve the numerical performance of PhyloMed. We have considered replacing the log-ratio subcomposition model in PhyloMed with the Beta-binomial model [41] and a distribution-free model for the composition count data [27]. The resulting global mediation tests based on these two alternative mediator models do not yield a better type I error control (Additional file 1: Table S3).

The assumption of no unmeasured confounding is critical in obtaining an unbiased estimate of the mediation effect and establishing the causal interpretation in mediation analysis. The correlation $\rho$ between the residuals of the mediator regression model and the outcome regression model is often used to quantify the magnitude of confounding bias on the mediation effect estimate with $\rho = 0$ implying no confounding bias [11]. Therefore, a common way to examine the sensitivity of finding to the violation of the assumption is to evaluate how the mediation effect estimate changes when $|\rho|$ deviates from zero [11]. Additional file 1: Fig. S9 displays the sensitivity analysis results for the identified mediating nodes highlighted in Fig. 4a and Additional file 1: Fig. S8 of the two real data analyses. The 90%

confidence interval of the estimated mediation effect covers zero when $|\rho| > 0.13$ in the mouse cecal study and $|\rho| > 0.27$ in the human gut study, which means the sign and significance of the estimated mediation effect remain unchanged if $|\rho|$ is not beyond those values. In our analyses, the sample residual correlations at the identified nodes are very close to zero with the absolute value smaller than $10^{-16}$, suggesting that we probably do not have a strong confounding bias in our analyses.

Power can be affected if the phylogenetic tree is misspecified. For instance, if the mediation taxa are clustered on the true tree but more scattered on the misspecified tree, signals on the internal nodes may become less condensed and more challenging to detect. Our simulation study shows that PhyloMed is still more powerful than the competing methods when the taxa are randomly scattered on the tree (Additional file 1: Fig. S4). If the phylogenetic tree is unavailable, PhyloMed can be applied to the taxonomy tree, which is almost always provided in microbiome data. Our R package has incorporated this option.

In this article, we focus on microbiome data from targeted amplicon sequencing rather than metagenomic sequencing. But the ideas of the method can be extended to metagenomics studies. For metagenomics data, multiple phylogenetic trees can be constructed, each based on sequencing data of a set of marker genes. The structures of these trees may differ from one another. One approach is to apply PhyloMed to the data derived from each marker gene and harmonize the results across different tree structures. This may increase the robustness against tree misspecification compared to the analysis using a single tree.

## Conclusions

Elucidating the causal role of the microbiome and identifying mediating microbial agents is an important step toward the development of strategies to manipulate the microbiome to augment desirable health outcomes. A growing number of microbiome studies in recent years have leveraged mediation analysis to discover the causal mediation effect of the microbiome. We develop PhyloMed to combat low statistical power in the microbiome mediation analysis, especially when the mediation signal is sparse and weak. PhyloMed framework builds upon a phylogeny-guided divide-and-conquer strategy to search for the mediation signals in high-dimensional microbial compositions. A new testing procedure is developed to solve the problem of conservativeness in testing the composite mediation null hypothesis. These features of PhyloMed are fundamentally different from existing methods and substantially boost the power of microbiome mediation tests. We envision that PhyloMed will accelerate the discovery of causal mediating microbes and facilitate biological interpretation in the context of phylogenetic trees. As a general methodology, PhyloMed can be applied to the mediation analysis of other high-dimensional compositional data. We have provided an efficient R package for the broad utility of the method.

## Methods

### PhyloMed framework

We consider a random sample of $n$ subjects measured on a set of OTUs. Suppose the OTUs are placed on the leaves of a rooted phylogenetic tree with $J$ internal nodes. For subject $i = 1, \ldots, n$, we let $(M_{ij}, 1 - M_{ij})$ be the subcomposition at the $j$th internal node. In the PhyloMed framework, instead of having a single model on the composition of all leaf-level OTUs, we build a collection of independent local models, each of

which is for a subcomposition on a particular internal node of the tree. This modeling approach is similar to a Polya tree process [42] and has been adopted by several methods for microbiome differential abundance testing [27, 28, 43]. Despite the independence of the subcomposition $M_{ij}$'s over internal nodes, this modeling approach allows a rich dependence structure among leaf-level OTUs [44].

We apply the log-ratio transformation to subcomposition and use the log-ratio variable (i.e., $\log\left(\frac{M_{ij}}{1-M_{ij}}\right)$) as the mediator. We need to deal with zeros in the log-ratio transformation. In each local mediation model at an internal node, we remove subjects with both components of the subcomposition being zero as they carry no information on the subcomposition. We add 0.5 to the counts aggregated to the left and right child nodes for the remaining subjects. Although adding a small value (pseudocount) is a simple and commonly used practice to avoid zeros in log transformation [45], the choice of pseudocount is arbitrary and there is no consensus on the optimal value. Studies have shown that the pseudocount approach can lead to biased normalization and the downstream data analysis can be sensitive to the choice of pseudocount [46, 47]. We conducted the sensitivity analysis to study how the choice of pseudocount may affect the performance of PhyloMed. Additional file 1: Table S4 shows the type I error and power results when we use the pseudocount of 0.1, 0.5, or 1. The type I error is controlled for all pseudocounts and the power is slightly higher with smaller pseudocount but the difference is negligible with the increased sample size. Results from real data analyses echo those in the simulation (Additional file 1: Table S5). The PhyloMed global test *p*-values from the analysis of the mouse cecal data with the pseudocount of 0.1, 0.5, or 1 are 0.064, 0.085, 0.126. The human gut microbiome data has a larger sample size and the corresponding PhyloMed global test *p*-values are 0.049, 0.047, and 0.057. These results demonstrate that PhyloMed is generally not sensitive to the choice of pseudocount. Alternative zero-handling approaches without relying on the arbitrary choice of pseudocount can also be applied. The universally best approach is still an open problem in the field and requires further research.

For each subject *i*, we let $T_i$ be the treatment variable, $Y_i$ be the outcome variable, and $\mathbf{X}_i$ be a set of confounders that may affect the treatment, mediator, and outcome. To represent the causal path diagram of the local mediation model at the *j*th internal node (Fig. 1), we apply the following regression models

$$E\left\{\log\left(\frac{M_{ij}}{1-M_{ij}}\right)\right\} = \boldsymbol{\alpha}_{jX}^{\mathrm{T}}\mathbf{X}_i + \alpha_j T_i, \tag{1}$$

$$g\{E(Y_i)\} = \boldsymbol{\beta}_{jX}^{\mathrm{T}}\mathbf{X}_i + \beta_{jT} T_i + \beta_j \log\left(\frac{M_{ij}}{1-M_{ij}}\right). \tag{2}$$

We omit the intercept term in both models as it can be absorbed into $\mathbf{X}_i$. Equation (1) represents a distribution-free linear regression model with the mean of the logit($M_{ij}$) depending on the treatment and confounders but the distribution of the subcomposition is not specified. Such a model is robust against the over-dispersion and outliers in the microbiome data. Equation (2) is a generalized linear regression with $g\{\cdot\}$ being

the link function depending on the type of outcome. As $M_{ij}$'s over internal nodes are independent, we can fit $J$ low-dimensional models $[Y_i \mid \mathbf{X}_i, T_i, \text{logit}(M_{ij})]$ as Eq. (2) as opposed to a large joint model $[Y_i \mid \mathbf{X}_i, T_i, \text{logit}(M_{i1}), \ldots, \text{logit}(M_{iJ})]$ for the purpose of hypothesis testing.

The potential outcomes framework [9–11] has established a series of identifiability assumptions such that models (1) and (2) lead to quantification of the causal mediation effect. A rigorous definition of causal mediation using potential outcomes framework is provided in Additional file 1: Note A.1 [7, 48]. An extension of the model to allow for the treatment-mediator interaction is described in Additional file 1: Note A.2.

### Composite null hypothesis tests in local mediation models

In each local mediation model, we are interested in testing whether the subcomposition at the $j$th internal node lies in the causal pathway from the treatment to the outcome. For both continuous and binary outcomes, the null and alternative hypotheses for this testing problem can be formulated as (details in Additional file 1: Note A)

$$H_0^j : \alpha_j \beta_j = 0 \quad \text{vs} \quad H_a^j : \alpha_j \beta_j \neq 0.$$

The null hypothesis can be equivalently expressed as the union of three disjoint null hypotheses

$$H_{00}^j : \alpha_j = \beta_j = 0,$$
$$H_{10}^j : \alpha_j \neq 0, \beta_j = 0,$$
$$H_{01}^j : \alpha_j = 0, \beta_j \neq 0.$$

The Sobel's test [24] and the joint significance test have been widely applied to test the mediation null hypothesis. Unfortunately, these tests have severely deflated type I error and lack power because they fail to take into account the composite nature of the null hypothesis [25, 31].

To address this issue, several new mediation tests were recently developed. JT-comp [49] assesses the product of two test statistics of exposure-mediator association and mediator-outcome association, the method proposed by Dai et al. [50] assesses the maximum of the two test $p$-values, and DACT [51] assesses a test statistic of a composite $p$-value. Dai's method and DACT estimate the proportion of the three types of nulls and assess their test statistics under different nulls. DACT employs Efron's empirical null inference framework [52] to further calibrate the $p$-value. In light of these methods, we propose a new testing procedure that handles the composite null hypothesis and produces a well-controlled type I error for multiple local mediation tests. Let $P_{\alpha_j}$ and $P_{\beta_j}$ denote the $p$-values for testing $\alpha_j = 0$ and $\beta_j = 0$, respectively. The $P_{\alpha_j}$ and $P_{\beta_j}$ are asymptotically uniformly distributed under their respective null hypotheses (i.e., $P_{\alpha_j}$ under $\alpha_j = 0$ and $P_{\beta_j}$ under $\beta_j = 0$) and are independent under the no unmeasured confounding assumptions. In PhyloMed, we adopt score statistics and obtain the observed $p$-values $p_{\alpha_j}$ and $p_{\beta_j}$ based on the reference asymptotic distribution or permutation (details in Additional file 1: Note B). The permutation $p$-value is more accurate than its asymptotic counterpart when the study sample size is small. We implement an adaptive

procedure to efficiently and accurately obtain the permutation *p*-values (Additional file [1]: Note B.3) [53].

We then define the mediation test statistic for $H_0^j$ as

$$P_{\max_j} = \max(P_{\alpha_j}, P_{\beta_j}).$$

The joint significance test also takes $P_{\max_j}$ as the mediation test statistic and determines statistical significance using the uniform distribution. In fact, $P_{\max_j}$ follows a mixture distribution with three components, each of which corresponds to one type of null hypothesis $H_{00}^j$, $H_{10}^j$, or $H_{01}^j$. Let $p_{\max_j}$ denote $\max(p_{\alpha_j}, p_{\beta_j})$, and $\pi_{00}$, $\pi_{10}$, and $\pi_{01}$ be the probabilities of the three null hypotheses among the $J$ local mediation models defined on the tree. The *p*-value of the mediation test in the *j*th local model is given by

$$
\begin{aligned}
&Pr(P_{\max_j} \leq p_{\max_j}) \\
=&\pi_{00}Pr(P_{\alpha_j} \leq p_{\max_j}, P_{\beta_j} \leq p_{\max_j} \mid H_{00}^j) \\
&+ \pi_{10}Pr(P_{\alpha_j} \leq p_{\max_j}, P_{\beta_j} \leq p_{\max_j} \mid H_{10}^j) + \pi_{01}Pr(P_{\alpha_j} \leq p_{\max_j}, P_{\beta_j} \leq p_{\max_j} \mid H_{01}^j) \\
=&\pi_{00}Pr(P_{\alpha_j} \leq p_{\max_j} \mid \alpha_j = 0)Pr(P_{\beta_j} \leq p_{\max_j} \mid \beta_j = 0) \\
&+ \pi_{10}Pr(P_{\beta_j} \leq p_{\max_j} \mid \beta_j = 0)Pr(P_{\alpha_j} \leq p_{\max_j} \mid \alpha_j \neq 0) \\
&+ \pi_{01}Pr(P_{\alpha_j} \leq p_{\max_j} \mid \alpha_j = 0)Pr(P_{\beta_j} \leq p_{\max_j} \mid \beta_j \neq 0) \\
=&\pi_{00}p_{\max_j}^2 + \pi_{10}p_{\max_j}Pr(P_{\alpha_j} \leq p_{\max_j} \mid \alpha_j \neq 0) + \pi_{01}p_{\max_j}Pr(P_{\beta_j} \leq p_{\max_j} \mid \beta_j \neq 0).
\end{aligned}
\tag{3}
$$

In this formula, we need to estimate three probabilities: $\pi_{00}$, $\pi_{10}$, and $\pi_{01}$, and two power functions evaluated at $p_{\max_j}$: $Pr(P_{\alpha_j} \leq p_{\max_j} \mid \alpha_j \neq 0)$ and $Pr(P_{\beta_j} \leq p_{\max_j} \mid \beta_j \neq 0)$.

We first employ the method proposed by Jin and Cai [54] to estimate $\pi_{0\bullet}$, the proportion of null $\alpha_j = 0$, using $p_{\alpha_j}$'s in all $J$ local mediation models. Specifically, we convert the $p_{\alpha_j}$ to Z-score $Z_{\alpha_j} = sign(\widehat{\alpha}_j) \times \Phi^{-1}(1 - p_{\alpha_j}/2)$, where $sign(\widehat{\alpha}_j)$ is the sign of the $\alpha_j$ estimate and $\Phi^{-1}(x)$ is the quantile function of the standard normal distribution. The empirical characteristic function and Fourier analysis are used to estimate the proportion of nulls. The empirical characteristic function is defined as

$$\varphi_J(t; Z_{\alpha_1}, \ldots, Z_{\alpha_J}) = \frac{1}{J}\sum_{j=1}^{J} e^{itZ_{\alpha_j}},$$

where $i = \sqrt{-1}$. The proportion of nulls can be consistently estimated as

$$\widehat{\pi}_{0\bullet} = \inf_{\left\{0 \leq t \leq \sqrt{\log(J)}\right\}} \left[ \int_{-1}^{1} (1 - |\xi|)\left\{ \mathrm{Re}(\varphi_J(t\xi; Z_{\alpha_1}, \ldots, Z_{\alpha_J}))e^{t^2\xi^2/2} \right\} d\xi \right],$$

where $\mathrm{Re}(x)$ denotes the real part of a complex number $x$. Similarly, we can obtain $\widehat{\pi}_{\bullet 0}$, the estimated proportion of null $\beta_j = 0$ using $p_{\beta_j}$'s across all local models. Then, under $H_0^j$, the estimates of $\pi_{00}$, $\pi_{10}$, and $\pi_{01}$ are $\widehat{\pi}_{00} = \widehat{\pi}_{0\bullet}\widehat{\pi}_{\bullet 0}/\widehat{\pi}_0$, $\widehat{\pi}_{10} = (1 - \widehat{\pi}_{0\bullet})\widehat{\pi}_{\bullet 0}/\widehat{\pi}_0$, and $\widehat{\pi}_{01} = \widehat{\pi}_{0\bullet}(1 - \widehat{\pi}_{\bullet 0})/\widehat{\pi}_0$, where $\widehat{\pi}_0 = \widehat{\pi}_{0\bullet} + \widehat{\pi}_{\bullet 0} - \widehat{\pi}_{0\bullet}\widehat{\pi}_{\bullet 0}$.

We also consider an alternative approach to estimate $(\pi_{00}, \pi_{10}, \pi_{01})$. In particular, we apply the method of Jin and Cai [54] to $p_{\max_j}$'s across all local models to estimate

the proportion of mediation null hypothesis (i.e., $\pi_0 = \pi_{00} + \pi_{10} + \pi_{01}$). We then estimate $\pi_{00}$, $\pi_{10}$, and $\pi_{01}$ as $\widehat{\pi}_{00} = (\widehat{\pi}_{0\bullet} + \widehat{\pi}_{\bullet 0} - \widehat{\pi}_0)/\widehat{\pi}_0$, $\widehat{\pi}_{10} = (\widehat{\pi}_0 - \widehat{\pi}_{0\bullet})/\widehat{\pi}_0$, and $\widehat{\pi}_{01} = (\widehat{\pi}_0 - \widehat{\pi}_{\bullet 0})/\widehat{\pi}_0$, where $\widehat{\pi}_{0\bullet}$ and $\widehat{\pi}_{\bullet 0}$ are the same estimates as before. The key difference between the two approaches is in the estimate of the proportion of mediation null hypothesis (i.e., $\pi_0$). The first approach uses the product of two proportions $\widehat{\pi}_0 = 1 - (1 - \widehat{\pi}_{0\bullet})(1 - \widehat{\pi}_{\bullet 0})$, and we refer to it as the "product" approach. The second approach directly estimate $\pi_0$ using $p_{\max_j}$, and we refer to it as the "maxp" approach. We evaluated the estimation bias and standard error of the two approaches in our simulation studies ("Methods" section "Simulation strategy"). The result shows that the two approaches are very similar (Additional file 1: Table S6). Results presented in the main text are based on the "product" approach. Our R package has incorporated both options.

For the two power functions, we employ the nonparametric estimates based on the Grenander estimator of the *p*-value density [55]. We show below the procedure for obtaining $\widehat{Pr}(P_{\alpha_j} \leq p_{\max_j} \mid \alpha_j \neq 0)$. Specifically, we use formula (9) in [55] to compute the Grenander estimator of the decreasing density at every $p_{\alpha_j}$, denoted by $\widehat{f}(p_{\alpha_j})$. Under $H_0^j$, $P_{\alpha_j}$ follows a mixture distribution composed of the uniform distribution (under $H_{00}^j$ and $H_{01}^j$) and the power function (under $H_{10}^j$). Therefore, we can obtain the conditional density estimate $\widehat{f}(p_{\alpha_j}|\alpha_j \neq 0)$ by solving the equation

$$\widehat{f}(p_{\alpha_j}) = \widehat{\pi}_{10}\widehat{f}(p_{\alpha_j} \mid \alpha_j \neq 0) + \widehat{\pi}_{00} + \widehat{\pi}_{01}.$$

We can then estimate $\widehat{Pr}(P_{\alpha_j} \leq p_{\max_j} \mid \alpha_j \neq 0)$ based on $\widehat{f}(p_{\alpha_j} \mid \alpha_j \neq 0)$ at all observed $p_{\alpha_j}$'s. We can obtain $\widehat{Pr}(P_{\beta_j} \leq p_{\max_j} \mid \beta_j \neq 0)$ using a similar estimation procedure. Finally, the mediation test *p*-value at the *j*th node can be estimated as

$$p_j = \widehat{\pi}_{00}p_{\max_j}^2 + \widehat{\pi}_{10}p_{\max_j}\widehat{Pr}(P_{\alpha_j} \leq p_{\max_j} \mid \alpha_j \neq 0) + \widehat{\pi}_{01}p_{\max_j}\widehat{Pr}(P_{\beta_j} \leq p_{\max_j} \mid \beta_j \neq 0).$$

In our numerical studies, we compare this mixture-distribution-based mediation test with the Sobel's test and the joint significance test (Additional file 1: Note C).

### Mediating subcomposition detection

We deduce below that the mediation test statistics $P_{\max_j}$'s of internal nodes under $H_0^j$ (referred to as null internal nodes) are asymptotically mutually independent as the sample size goes to infinity.

Under $H_0^j$, the two power function probabilities $Pr(P_{\alpha_j} \leq t_j \mid \alpha_j \neq 0)$ and $Pr(P_{\beta_j} \leq t_j \mid \beta_j \neq 0)$ converge to 1 for a constant $t_j$ when the sample size goes to infinity. Therefore, based on formula (3), $Pr(P_{\max_j} \leq t_j)$ can be approximated by

$\pi_{00}Pr(P_{\alpha_j} \leq t_j \mid \alpha_j = 0)Pr(P_{\beta_j} \leq t_j \mid \beta_j = 0) + \pi_{10}Pr(P_{\beta_j} \leq t_j \mid \beta_j = 0) + \pi_{01}Pr(P_{\alpha_j} \leq t_j \mid \alpha_j = 0) = \pi_{00}t_j^2 + \pi_{10}t_j + \pi_{01}t_j$ .

Clearly, the formula only involves probabilities of $P_{\alpha_j}$ and $P_{\beta_j}$ under their respective nulls $\alpha_j = 0$ and $\beta_j = 0$. Indeed, the *p*-value under the null is stochastically larger than the *p*-value under the alternative. At the null internal node *j* (at least one of the $\alpha_j$ and $\beta_j$ is zero), the $P_{\max_j}$ converges to the *p*-value ($P_{\alpha_j}$ or $P_{\beta_j}$) under the null ($\alpha_j = 0$ or $\beta_j = 0$) as the sample size goes to infinity. Moreover, $P_{\alpha_j}$'s are asymptotically independent over the nodes under $\alpha_j = 0$ and $P_{\beta_j}$'s are asymptotically independent over the nodes under $\beta_j = 0$ because subcompositions $M_j$'s over internal nodes are modeled as independent variables. Hence, $P_{\max_j}$'s over null internal nodes are also asymptotically independent.

When there is no mediating node in the tree (global mediation null setting), the ($\widehat{\pi}_{00}$, $\widehat{\pi}_{10}$, and $\widehat{\pi}_{01}$) converge to the true values and yield uniformly distributed mediation *p*-values under the large sample size. In the presence of mediating taxa, the $\widehat{\pi}_{00}$ underestimates the true $\pi_{00}$, and the $\widehat{\pi}_{10}$ and $\widehat{\pi}_{01}$ overestimate their corresponding parameter values. This is because the estimate of the proportion of mediation null hypothesis (i.e., $\widehat{\pi}_0$ estimated using "product" or "maxp" approach described in the last section) is inherently higher than the true value in the presence of mediating taxa, resulting in deflated $\widehat{\pi}_{00}$. This has been demonstrated in the simulation where the bias of $\widehat{\pi}_{00}$ is always negative under the settings $|S_\alpha| = |S_\beta| = 3$ or 6 in Additional file 1: Table S6. Consequently, the mediation test *p*-value at the null internal node is superuniform in the presence of mediating taxa because $\widehat{\pi}_{00}t_j^2 + \widehat{\pi}_{10}t_j + \widehat{\pi}_{01}t_j < \pi_{00}t_j^2 + \pi_{10}t_j + \pi_{01}t_j$ when $\widehat{\pi}_{00} < \pi_{00}$.

Given the properties of mediation *p*-values shown above, to control FDR in multiple testing, we can apply the standard BH procedure [29] to identify a collection of nodes with significant mediation effects on the phylogenetic tree.

## Global mediation test

We can combine all subcomposition mediation test *p*-values to test the global mediation null hypothesis that there is no mediation effect in any of the internal nodes (i.e., $H_0 : \cap_{j=1}^{J} H_0^j$). Here, we employ the HMP method [30]. Specifically, the weighted harmonic mean of the subcomposition mediation test *p*-values $p_1, \ldots, p_J$ is defined as

$$\mathring{p} = \frac{\sum_{j=1}^{J} w_j}{\sum_{j=1}^{J} w_j p_j},$$

where $w_j$'s are weights that sum to 1 and we set $w_j = 1/J$ by default. The global test *p*-value can be obtained by calculating the tail probability from the $\mathring{p}$'s null distribution approximation [30].

## Simulation strategy

To resemble reality, we used a real microbiome dataset [32] as a basis for the simulation. The data contained microbiome samples from 900 healthy subjects. We chose this dataset as our basis because of its large sample size. The data from a healthy cohort is representative of the distribution of microbiome without major perturbation. In each round of the simulation, we randomly sampled $n = 50$ or 200 subjects out of the 900 and divided them into two equal-sized treatment and control groups ($T_i = 1$   or   0). We used the top 100 most abundant OTUs and the associated phylogenetic tree with 99 internal nodes. Let $\mathcal{S}_\alpha$ and $\mathcal{S}_\beta$ denote the set of treatment-associated and outcome-associated OTUs, respectively, and $|\mathcal{S}|$ be the number of elements in $\mathcal{S}$. Under the null of no mediation effect, these two sets of OTUs do not overlap and we consider five combinations of ($|\mathcal{S}_\alpha|, |\mathcal{S}_\beta|$) values: (0, 0), (3, 0), (6, 0), (0, 3), (0, 6). Different values represent different mixtures of nulls $H_{00}$, $H_{10}$, and $H_{01}$: (0, 0) pertains to the setting where all local models are under $H_{00}$; (3 or 6, 0) pertains to the setting where some local models are under $H_{00}$ and others are under $H_{10}$; (0, 3 or 6) pertains to the setting where some local models are under $H_{00}$ and others are under $H_{01}$. Under the alternative, we let $\mathcal{S}_\alpha = \mathcal{S}_\beta$ and both sets index mediating OTUs that are associated with the treatment and the

outcome. We considered 3 or 6 mediating taxa clustered on the tree. In the case of 3 mediating taxa, we randomly selected a clade with three descendant OTUs and assigned them as mediators. In the case of 6 mediating taxa, we randomly selected two clades with three descendant taxa in each and assigned the six OTUs as mediators. We generated 2000 simulated datasets for each setting.

To perturb the abundance of each treatment-associated OTU $k \in \mathcal{S}_\alpha$, we randomly decide if we change the abundance of the OTU in the treatment group or the control group with equal probability. For each subject $i$ in the chosen group, we increased its abundance by adding a random count sampled from $Binomial(N_i, Af_k)$, where $N_i$ is the sequencing depth of subject $i$, $f_k$ is the average observed proportion of OTU $k$ across all the samples in the data, $A$ controls the strength of the treatment-mediator association, and we set $A = 0.5$. To simulate the outcome, we used the log-contrast regression model [33] that imposes a zero-sum constraint on the association coefficients to account for the compositional nature of the covariates. In our simulation, we considered both continuous and binary outcomes. For the continuous outcome, we generated data from the linear log-contrast regression model

$$Y_i = \beta_T T_i + \sum_{k \in \mathcal{S}_\beta} \beta_k \log(f_{ik}) + \epsilon_i, \text{ subject to } \sum_{k \in \mathcal{S}_\beta} \beta_k = 0,$$

where $f_{ik}$ is the observed proportion of OTU $k$ in subject $i$ and $\epsilon_i$ is the zero-mean normal error. For the binary outcome, we generated data from the logistic log-contrast regression model

$$\text{logit}\{Pr(Y_i = 1)\} = \beta_T T_i + \sum_{k \in \mathcal{S}_\beta} \beta_k \log(f_{ik}), \text{ subject to } \sum_{k \in \mathcal{S}_\beta} \beta_k = 0.$$

In these models, the coefficient $\beta_T$ was sampled from $Uniform(0, 1)$ and $\beta_k$'s were sampled from $Uniform(0, B)$, where $B$ controls the strength of the mediator-outcome association, and we set $B = 0.5$. The values of $\beta_k$'s were centered such that the zero-sum constraint was satisfied.

Additional simulation study was performed to evaluate the power of the PhyloMed global test with more taxa (especially the rare taxa). We include all OTUs in the basis dataset (rather than the top 100 in the default setting). In each simulated data, we retained OTUs with at least one non-zero observation. With the sample size of 200, the median number of retained OTUs across replicates of simulation is 422.

To evaluate the performance of different methods when the mediation signals are denser and the mediating OTUs are randomly scattered rather than clustered on the tree, we conducted another set of simulations with the continuous outcome. Specifically, we considered 3, 6, and 15 randomly scattered mediating OTUs. We also lowered the mediation effect size by decreasing $A$ to 0.001 to evaluate how PhyloMed performs under challenging cases.

### Software tools
MedTest: R package "MedTest" version 1.1, https://github.com/jchen1981/MedTest
MODIMA: R script "modima.R", https://github.com/alekseyenko/MODIMA
LDM-med: R package "LDM" version 5.0, https://github.com/yijuanhu/LDM

CMM: R package "ccmm" version 1.0 on CRAN

microHIMA: R package "HIMA" version 2.0.1, https://github.com/YinanZheng/HIMA

Sensitivity analysis: R package "mediation" version 4.5.0 on CRAN

## Supplementary Information

> **Additional file 1.** Note A: Definition of mediation effect via potential outcomes framework and identifiability assumptions; Note B: Details on obtaining $p_\alpha$ and $p_\beta$; Note C: Sobel's test and joint significance test; Table S1-S6; Figure S1-S9.
>
> **Additional file 2.** Review history.

### Acknowledgements
The authors thank editors and three anonymous reviewers for insightful comments and suggestions that helped improve the quality of the manuscript.

### Peer review information

### Review history
The review history is available as Additional file 2.

### Authors' contributions
ZZT oversaw the study. The theory underlying PhyloMed was conceived of and developed by ZZT, with contributions from GC and QH. QH performed simulation studies, real data analyses and developed miMediation R package. ZZT wrote the first version of the manuscript. QH and GC also contributed to the writing. The authors read and approved the final manuscript.

### Authors' Twitter handles
Twitter handles: @tangzz_lab (Zheng-Zheng Tang).

### Funding
This work was supported by the NIH grant R01GM140464, NSF grant DMS-2054346, University of Wisconsin-Madison Center for Demography of Health and Aging (CDHA) pilot grant, and Data Science Initiative Award provided by the University of Wisconsin-Madison Office of the Chancellor and the Vice Chancellor for Research and Graduate Education with funding from the Wisconsin Alumni Research Foundation.

### Availability of data and materials
The proposed method is implemented in the R package miMediation, publicly available at Github: https://github.com/KiRinHong/miMediation [56]. The scripts generating reported results are deposited at Zenodo: https://doi.org/10.5281/zenodo.7443578 [57]. Both repositories are licensed under the open-source GPL-3.0. The mouse cecal microbiome data are available at the NLM repository under the accession ID: BioProject 168618 [34]. The human gut microbiome data are available at the Qiita repository under the accession ID: 1011 [35].

## Declarations

### Ethics approval and consent to participate
Not applicable.

### Consent for publication
Not applicable.

### Competing interests
The authors declare that they have no competing interests.

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

## 

