## [**Additional file 2.** Review history. · Genome Biology]

Review History

First round of review

Reviewer 1

Were you able to assess all statistics in the manuscript, including the appropriateness of statistical tests used? Yes: This ms proposes a new statistical method, so the entire review is about the statistics.

Were you able to directly test the methods? No.

Comments to author:

This manuscript proposed a new method for testing the mediation effect of each subcomposition (i.e., internal node) on a phylogenetic tree, the p-values of which are aggregated via the harmonic mean method for testing the overall mediation effect of the entire community. Power can be gained by testing the ancestor nodes of mediating taxa, at which mediation signals were condensed.

Major Comments:

A major limitation with this method is that the taxa at the lowest level of the tree cannot be tested individually. This seems to be a major limitation, and in fact forces some unattractive assumptions around the requirement that there must be leaf taxa that both contribute to the mediation effect and the true association. This is illustrated in the somewhat paradoxical results in Supplementary Figure 6 (final column) and discussed in the paragraph that starts on line 321. This limitation (and any advantages that are achieved by the assumptions that lead to this limitation) should be discussed earlier in the main texts and its implications discussed in greater length.

The theory was developed under the global null hypothesis of no mediating taxa while it is more relevant to study the theory in the presence of some mediating taxa. Specifically, the authors deduced that the PhyloMed local mediation test p-values are asymptotically mutually independent and uniformly distributed under the global null hypothesis that no mediation effect in any of the internal nodes. However, this does not translate into a uniform (or superuniform) distribution of p-values at null taxa in the presence of other taxa that are mediating taxa, which is required by the BH procedure.

All simulation studies are based on a “spike-in” model in which, to perturb treatment associated OTU abundance, a random number of counts was added to the subjects in the treatment group. This has the effect of making the library sizes in the treatment group systematically higher than those in the control group, which is often not the case in real data (library size is usually an artifact of sequencing). Other simulation models that more accurately represent experimental data should be considered to assess robustness of the proposed method.

Since the mediation effects were simulated in terms of ratios of relative abundances, to ensure a fair comparison, MedTest and MODIMA should be applied to the Aitchison distance, which is the Euclidean distance applied to the clr-transformed data.

All simulations are based on (the top 100) common taxa. It is unclear how the proposed method would perform at (relatively) rare taxa and how to calibrate the taxa for which the method performs well.

It should be justified why a dataset that was neither used in real data analysis nor a (well known) benchmarking dataset was used for simulation studies.

The method by Yue and Hu (2022, Bioinformatics) can also detect the overall mediation effect of the entire microbial community and pinpoint individual taxa. This reference should be cited, and the results of that method should be included in comparisons.

On page 3, it was stated that “Other mediation analysis methods... select mediating taxa and some provide global tests of the overall mediation effect at the community level”. More comparisons between these methods and the proposed method, especially those that motivated the development of the new method, should be stated.

The authors pointed out the uncertainty in the phylogenetic tree constructed from metagenomic data. In fact, the accuracy of the phylogenetic tree constructed from 16S data is also questionable. The consequence of using an inaccurate tree should be discussed.

The pseudocount approach to handling zeros has been found to be potentially problematic, as in the presence of mediating taxa, there may be differential proportions of zeros associated with either the exposure or the outcome and hence differential effects of adding the pseudocount. This potential problem should be discussed; results using different pseudocounts could be compared to indicate how strongly the results depend on the choice of pseudocount.

The type I error rates for testing the global null are close to the nominal level but the empirical FDR values (Supplementary Table S1) are quite conservative. It would be interesting to discuss this discrepancy, and reconcile these seemingly contradictory results if possible.

At the top of page 18, why the independence of the exposure-microbe association and the microbe-outcome association can be assumed in the estimation of π_{00} , π_{01} , and π_{10} ? There should be arbitrary proportions of the three null hypotheses.

Minor Comments:

More detail should be given to the most important competing methods, Sobel’s test and joint significance test. Also, it is unclear how these tests were applied to obtain the p-value for testing the subcomposition mediation effect in each local model.

Please be aware that for some readers, it is very difficult to distinguish the red and purple circles in Supplementary Figure 6. You may wish to consider using shapes as well as colors.

Reviewer 2

Were you able to assess all statistics in the manuscript, including the appropriateness of statistical tests used? Yes: Yes I am able to assess all statistics.

Were you able to directly test the methods? No.

Comments to author:

The authors proposed a phylogeny-based method, PhyloMed, to identify human microbes which act as mediators between treatments (or exposures) and health outcomes. PhyloMed uses local mediation models on the internal nodes of the phylogenetic tree. The manuscript is well written. The proposed method outperforms existing methods in simulation studies. I have a couple concerns and hope the authors can clarify and elaborate further.

- 1) The authors claimed that the local models are independent of each other. However, there is a hierarchical order of the phylogenetic tree, the internal nodes are children nodes of other nodes. How could the tests be independent of each other?
- 2) For the simulation studies, the number of mediating microbes is very small. Is this close to real world? When the number of mediating microbes is large, will FDR be still controlled?
- 3) The authors did the test for each internal node separately. How will you evaluate the joint direct and indirect mediation effects?
- 4) For application purpose, using the proposed method, under what situations, leaf nodes are not identified as mediators but the internal nodes are mediators?

Reviewer 3

Were you able to assess all statistics in the manuscript, including the appropriateness of statistical tests used? Yes.

Were you able to directly test the methods? Yes.

Comments to author:

The authors proposed a phylogeny-based mediation analysis method PhyloMed to account for the compositionality and high-dimensionality of microbiome data, as well as the composite nature of null hypotheses.

Major comments:

1. To test the mediation effect, the authors assumed that the microbiome data follows assumptions of no unmeasured confounders. However, real microbiome data is really complicated and these assumptions might be violated, how to deal with the unmeasured confounders?
2. As the authors mentioned in the manuscript, Sobel's test and joint significance test tend to produce conservative type I error rate. In order to deal with such issue, several method have been proposed[1-2]. In particular, Liu et al [2] develops DACT to control Type I error rate by estimating the proportions of the three null cases. The authors should claim the differences between PhyloMed and DACT and compare it with DACT in the comparative study.
3. The proposed method PhyloMed is a kind of community level test (global test) by combining the testing results on each internal node. Could the authors provide the testing results on the taxon level using the p values on the internal node? If possible, the authors should provide some simulation studies to compare PhyloMed with other methods[3-6].
4. The authors developed PhyloMed to deal with the compositional and high-dimensional nature of microbiome incorporating phylogenetic tree, while addressing the composite null hypotheses.

What if the phylogenetic tree was not provided or misspecified, could PhyloMed still control the type I error rate?

- [1] James Y. Dai, Janet L. Stanford & Michael LeBlanc. A Multiple-Testing Procedure for High-Dimensional Mediation Hypotheses, *Journal of the American Statistical Association*. 2022;117(537):198-213.
- [2] Zhonghua Liu, Jincheng Shen, Richard Barfield, Joel Schwartz, Andrea A. Baccarelli & Xihong Lin. Large-Scale Hypothesis Testing for Causal Mediation Effects with Applications in Genome-wide Epigenetic Studies, *Journal of the American Statistical Association*. 2022;117(537):67-81.
- [3] SohnMB, LiH. Compositionalmediationanalysisformicrobiomestudies. *The Annals of Applied Statistics*. 2019;13(1):661-681.
- [4] Wang C, Hu J, Blaser MJ, et al. Estimating and testing the microbial causal mediation effect with high-dimensional and compositional micro- biome data. *Bioinformatics*. 2020;36(2):347-355.
- [5] Zhang H, Chen J, Feng Y, et al. Mediation effect selection in high- dimensional and compositional microbiome data. *Statistics in Medicine*. 2021;40(4):885-896.
- [6] Zhang H, Chen J, Li Z, et al. Testing for mediation effect with application to human microbiome data. *Statistics in Biosciences*. 2021;13(2):313-328.

Response to Reviews

We thank the reviewers and the Editor for their consideration and constructive comments on our manuscript “PhyloMed: a phylogeny-based test of mediation effect in microbiome” (GBIO-D-22-00997). We have performed new numerical studies and clarified several points to address reviewers’ comments in the detailed responses below. We also summarized the broad utility and novelty of PhyloMed at the end of this letter and in the Conclusion section of the revised manuscript. We believe that the new manuscript is substantially improved.

Reviewer #1:

This manuscript proposed a new method for testing the mediation effect of each subcomposition (i.e., internal node) on a phylogenetic tree, the p-values of which are aggregated via the harmonic mean method for testing the overall mediation effect of the entire community. Power can be gained by testing the ancestor nodes of mediating taxa, at which mediation signals were condensed.

Many thanks for the reviewer’s time and thoughtful suggestions about our work.

Major Comments:

A major limitation with this method is that the taxa at the lowest level of the tree cannot be tested individually. This seems to be a major limitation, and in fact forces some unattractive assumptions around the requirement that there must be leaf taxa that both contribute to the mediation effect and the true association. This is illustrated in the somewhat paradoxical results in Supplementary Figure 6 (final column) and discussed in the paragraph that starts on line 321. This limitation (and any advantages that are achieved by the assumptions that lead to this limitation) should be discussed earlier in the main texts and its implications discussed in greater length.

Response: Mediation signal consists of two elements: treatment-mediator association and mediator-outcome association (conditional on the treatment effect). The fundamental problem we would like to call readers’ attention to is the separation of the two elements. The assumption essentially states that “there is no complete separation of the two elements”. We agree that this is an important issue for interpreting the results from microbiome mediation analysis and we address this comment from the three perspectives listed below. We hope to clarify that (1) this assumption is for the interpretation of the results and not required by the validity of the method, (2) this problem universally exists for mediation analysis of compositional mediators, even for the methods applied to taxa in one taxonomic level (e.g. leaf-level taxa).

1. The mediation effect at the ancestor taxon can be formed by aggregating lower-level taxa associated with only the treatment and lower-level taxa associated with only the outcome (i.e., the separation of two elements demonstrated in the last column of Supplementary Fig. 6 in the initial manuscript or Supplementary Fig. 9 in the revised manuscript). This is conceptually different from the false positives or paradoxical results as the mediation effect indeed exist at the ancestor node and it is not possible to see the separation of the two elements at the leaf-level taxa by only testing the ancestor node (PhyloMed does not test leaf-level taxa). Hence, the assumption is used to extend the interpretation of the significant mediation signal (discovered in PhyloMed or any other methods) from upper level taxa to the lower level, but the validity of the method does not rely on this assumption.

2. The leaf-level taxa are also usually formed by binning/aggregating when processing the sequencing reads. For example, in the 16S rRNA sequencing data, each OUT/ASV represents a group of bacteria whose genome homology is somewhere between strain and species. Moreover, people commonly work with genera from aggregating species in 16S rRNA sequencing data for better data quality. Suppose a method is applied to the leaf-level taxa and identifies a mediating leaf-level taxon. This assumption is also needed if one wants to state that the mediation signal exists at an even lower level (e.g. strain).
3. Suppose we only want to identify mediating taxa at the leaf level. The two elements are still subject to separation when we analyze individual taxa at one taxonomic level because of the compositional nature of the microbiome mediator. The relative abundances of all taxa are linked because of the unit-sum constraint of the proportions across taxa: changing the absolute abundance of one taxon would shift the relative abundance of all other taxa. Consequently, if a taxon is identified as a mediator using its relative abundance (i.e., observed proportion in the composition), the two elements contributing to the mediation signal may come from entirely different sets of taxa. To handle compositional data, many methods assume that the mediation signal is sparse and apply different transformations to the relative abundance (e.g., additive log-ratio, isometric log-ratio, see Background) in defining the mediators and the mediation model. The mediation effects are "relative effects" defined and interpreted in the context of a particular transformed composition adopted by a method. The assumption of no complete separation of the two elements is needed if one wants to interpret the identified mediation signal in that context as having the true mediation effect (i.e., no complete separation of the two elements in the underlying absolute abundance data). Therefore, we are hesitant to agree with the reviewer's comment that not being able to test individual taxa is the major limitation of PhyloMed because, without absolute abundance data, no methods can estimate the true "mediation effect" at individual taxa. Without the assumption, the interpretation of the mediation signal identified by any method is limited to a specific way the method defines the mediators from the microbial composition. This is probably a reason why many mediation methods for microbiome data focus more on hypothesis testing than estimation. The test results provide a scan of high-dimensional microbial composition and generate candidates for downstream validation studies and mechanistic experiments. Different methods provide different ways to reduce search space and find candidates. PhyloMed, empowered by the phylogeny-guided divide-and-conquer strategy and efficient local mediation test, provides a novel and efficient way to search for candidates. Even though PhyloMed does not directly test leaf-level taxa, the number of candidates under the PhyloMed-identified internal node is greatly smaller than all the leaf-level taxa.

The above points are consolidated into two paragraphs and moved at the end of Discussion.

The theory was developed under the global null hypothesis of no mediating taxa while it is more relevant to study the theory in the presence of some mediating taxa. Specifically, the authors deduced that the PhyloMed local mediation test p-values are asymptotically mutually independent and uniformly distributed under the global null hypothesis that no mediation effect in any of the internal nodes. However, this does not translate into a uniform (or superuniform) distribution of p-values at null taxa in the presence of other taxa that are mediating taxa, which is required by the BH procedure.

Response: We thank the reviewer for the helpful suggestion. In the revision, we studied the behavior of the mediation test p-value at the null internal node in the presence of mediating taxa. In this case, we showed that the mediation p-value at the null internal node is superuniform. To better organize the materials, in the revision, we divided the Methods section “Global mediation test and mediating subcomposition detection” into two sections: one is “Mediating subcomposition detection” and the other is “Global mediation test” and the proof is placed in the former section (the first paragraph on page 23), which concerns the setting where some internal nodes are under the null and others are under the alternative.

All simulation studies are based on a “spike-in” model in which, to perturb treatment associated OTU abundance, a random number of counts was added to the subjects in the treatment group. This has the effect of making the library sizes in the treatment group systematically higher than those in the control group, which is often not the case in real data (library size is usually an artifact of sequencing). Other simulation models that more accurately represent experimental data should be considered to assess robustness of the proposed method.

Response: In our revision, we have changed how to perturb treatment-associated OTUs. In particular, for each treatment-associated OTU, we randomly decide whether we add the random count to the treatment group or the control group with equal probability. By implementing this change, the treatment effect on the OTU can be positive or negative, and the average library sizes in the two groups are similar. This simulation strategy better represents real data. We have updated all simulation results using this new strategy.

Since the mediation effects were simulated in terms of ratios of relative abundances, to ensure a fair comparison, MedTest and MODIMA should be applied to the Aitchison distance, which is the Euclidean distance applied to the clr-transformed data.

Response: In MedTest and MODIMA tests, we added the Aitchison distance and repeated all numerical studies. This distance slightly improves the power of MedTest and MODIMA in some simulation settings, but their power remains lower than PhyloMed. We have updated all simulation and real data application results in the revision.

All simulations are based on (the top 100) common taxa. It is unclear how the proposed method would perform at (relatively) rare taxa and how to calibrate the taxa for which the method performs well.

Response: We used the top 100 taxa because they make up 97.8% of the full composition and already contain many rare taxa (e.g., 13 taxa have more than 80% of their observations being zero). However, we agree with the reviewer that it would be interesting to evaluate the setting where we have more taxa, especially the rare ones. We have performed an additional simulation study. In particular, we included all 819 OTUs in the basis dataset. In each simulated data, we retained OTUs with at least one non-zero observation. With the sample size of 200, the median number of retained OTUs across replicates of simulation is 422. In this case, all methods become less powerful (comparing Fig. 3 with Supplementary Fig. 2), but the power of PhyloMed remains the highest. The CMM method cannot be evaluated when we include all OTUs because CMM fails to converge when the sample size is smaller than the number of taxa. We have included this additional simulation in the revision (the first paragraph on page 8 and the second to the last paragraph on page 25).

It should be justified why a dataset that was neither used in real data analysis nor a (well known) benchmarking dataset was used for simulation studies.

Response: The main reason we use that particular dataset as the basis for the simulation is because of the large sample size (900 subjects). We need to generate 2000 replicates of simulated datasets with $n=50$ or 200 individuals. We need a large enough pool in the basis dataset to sample from so that we have enough variability over replicates. The two datasets in the real data analysis have too small of a sample size to serve as a sampling pool for simulation. The other consideration is that the basis dataset contains samples from a healthy cohort, which are better representative of the distribution of microbiome without major perturbation than datasets from disease cohorts. The dataset is the largest data from a healthy cohort available to us. We have added this justification in the first paragraph on page 24.

The method by Yue and Hu (2022, Bioinformatics) can also detect the overall mediation effect of the entire microbial community and pinpoint individual taxa. This reference should be cited, and the results of that method should be included in comparisons.

Response: We thank the reviewer for calling out this recent paper that developed the LDM-med method. We have added this method in the discussion of existing methods (see Background) and in our numerical studies. The type I error of the LDM-med global test shows a deflation pattern similar to MedTest and MODIMA (Fig. 2 and Supplementary Fig. 1). When the mediation effect is large, the power of the LDM-med global test is higher than MedTest and MODIMA in most settings but lower than the PhyloMed global test (Fig. 3 and Supplementary Figs 2-3). When the mediation effect is small, the LDM-med test has much lower power than other methods (Supplementary Fig. 3). In real data applications, LDM-med cannot identify any mediating taxa in the two datasets and the global test p-values are not significant. These additional numerical studies have been included in the revision.

On page 3, it was stated that “Other mediation analysis methods... select mediating taxa and some provide global tests of the overall mediation effect at the community level”. More comparisons between these methods and the proposed method, especially those that motivated the development of the new method, should be stated.

Response: We have expanded our description of these existing regularized methods and included in the following paragraph in Background.

“Other mediation analysis methods for microbiome data assume sparse mediation effects and estimate the effects via regularization. They aim to select mediating taxa and some provide global tests of the overall mediation effect at the community level. These methods apply different treatments to handle compositional data. CMM uses the composition operators to define the mediation model with the parameters interpreted under the additive log-ratio transformation; microHIMA uses the isometric log-ratio to transform the relative abundance to variables in the Euclidean space; SparseMCMM uses the Dirichlet regression to model microbial compositions. These methods also employ different mediation tests. CMM uses a Sobel-type test; microHIMA uses a joint-significance-type test. SparseMCMM includes two tests: one uses the overall mediation effect estimate as the test statistic and the other uses the sum of squares of the component-wise mediation effect estimates as the test statistic, both of which have conservative control of type I error reported in the original paper”

In Conclusions, we have summarized the main motivation and novelty of PhyloMed in the following text:

“We develop PhyloMed to combat low statistical power in the microbiome mediation analysis, especially when the mediation signal is sparse and weak. PhyloMed framework builds upon a phylogeny-guided divide-and-conquer strategy to search for the mediation signals in high-dimensional microbial compositions. A new testing procedure is developed to solve the problem of conservativeness in testing the composite mediation null hypothesis. These features of PhyloMed are fundamentally different from existing methods and substantially boost the power of microbiome mediation tests.”

The authors pointed out the uncertainty in the phylogenetic tree constructed from metagenomic data. In fact, the accuracy of the phylogenetic tree constructed from 16S data is also questionable. The consequence of using an inaccurate tree should be discussed.

Response: In the revision, we have discussed the consequence of tree misspecification in the text below. This is provided in the second paragraph on page 13.

“Power can be affected if the tree is misspecified. For instance, if the mediation taxa are clustered on the true tree but more scattered on the misspecified tree, signals on the internal nodes may become less condensed and more challenging to detect. Our simulation study shows that PhyloMed is still more powerful than the competing methods when the taxa are randomly scattered on the tree (Supplementary Fig. 3).”

The pseudocount approach to handling zeros has been found to be potentially problematic, as in the presence of mediating taxa, there may be differential proportions of zeros associated with either the exposure or the outcome and hence differential effects of adding the pseudocount. This potential problem should be discussed; results using different pseudocounts could be compared to indicate how strongly the results depend on the choice of pseudocount.

Response: We agree that the pseudocount approach is not problem-free. We have discussed the potential problems of the pseudocount approach and conducted additional numerical studies to evaluate how strongly the results depend on the choice of pseudocount. The discussion and new results are provided in a paragraph below (on page 16-17 in the revision).

“Although adding a small value (pseudocount) is a simple and commonly used practice to avoid zeros in log transformation, the choice of pseudocount is arbitrary and there is no clear consensus on the optimal value. Studies have shown that the pseudocount approach can lead to biased normalization and the downstream data analysis can be sensitive to the choice of pseudocount. We conducted the sensitivity analysis to study how the choice of pseudocount might affect the performance of PhyloMed. Supplementary Table 4 shows the type I error and power results when we use the pseudocount of 0.1, 0.5, or 1. The type I error is controlled for all pseudocounts and the power is slightly higher with smaller pseudocount but the difference is negligible with the increased sample size. Results from real data analyses echo those in the simulation (Supplementary Table 5). The PhyloMed global test p-values from the analysis of the mouse cecal data with the pseudocount of 0.1, 0.5, or 1 are 0.064, 0.085, 0.12. The human gut microbiome data has a larger sample size and the corresponding PhyloMed global test p-values are 0.049, 0.047, and 0.057. These results demonstrate that PhyloMed is generally not sensitive to the choice of pseudocount. Alternative

zero-handling approaches without relying on the arbitrary choice of pseudocount can also be applied. The universally best approach is still an open problem in the field and requires further research.”

The type I error rates for testing the global null are close to the nominal level but the empirical FDR values (Supplementary Table S1) are quite conservative. It would be interesting to discuss this discrepancy, and reconcile these seemingly contradictory results if possible.

Response: We agree that the empirical FDR should be close to the type I error when there is no mediation signal in any internal node (i.e., under the global null). In the presence of the mediation signal, the empirical FDR can be affected by the signal pattern (e.g., strength, density). The BH method usually controls the FDR at a more stringent level than the target FDR. We have performed additional simulation studies of empirical FDR by varying mediation signal strength and density (Supplementary Fig. 4). We can see that the empirical FDR is smaller than the target level as the signal get stronger and denser (added in the last paragraph on page 8). Many modern FDR control methods can be adopted to replace BH in detecting mediating nodes but how much the modern FDR methods improve the classic BH method depends on many factors (e.g., the informativeness of the auxiliary/structural information, signal density) [1]. A comprehensive review and comparison of these different methods are beyond the scope of this paper and we leave this interesting topic for further research.

[1] Korthauer, Keegan, et al. A practical guide to methods controlling false discoveries in computational biology. *Genome Biology* 20.1 (2019): 1-21.

At the top of page 18, why the independence of the exposure-microbe association and the microbe-outcome association can be assumed in the estimation of π_{00} , π_{01} , and π_{10} ? There should be arbitrary proportions of the three null hypotheses.

Response: In each local mediation model, the exposure-mediator association test and the mediator-outcome association test are independent because of the factorization of the likelihood for α_j and the likelihood for β_j . However, we fully appreciate the reviewer’s point that the proportions of the three null hypotheses should be arbitrary. Therefore, we have considered an alternative way to estimate $(\pi_{00}, \pi_{10}, \pi_{01})$. In particular, we estimate the proportion of the mediation null hypothesis (i.e. $\pi_0 = \pi_{00} + \pi_{10} + \pi_{01}$) using $pmax_j = \max(p_{\alpha_j}, p_{\beta_j})$ over all internal nodes. We then estimate the $(\pi_{00}, \pi_{10}, \pi_{01})$ using formulas: $\hat{\pi}_{00} = (\hat{\pi}_0 + \hat{\pi}_{\cdot 0} - \hat{\pi}_0)/\hat{\pi}_0$, $\hat{\pi}_{10} = (\hat{\pi}_0 - \hat{\pi}_{\cdot 0})/\hat{\pi}_0$ and $\hat{\pi}_{01} = (\hat{\pi}_0 - \hat{\pi}_{0\cdot})/\hat{\pi}_0$

The simulation results show that the estimates from the two approaches yield very similar bias and standard error (Supplementary Table 6). The numerical studies still use the original approach but our R package has incorporated both options. We have added these additional materials in the second paragraph on page 21.

Minor Comments:

More detail should be given to the most important competing methods, Sobel's test and joint significance test. Also, it is unclear how these tests were applied to obtain the p-value for testing the subcomposition mediation effect in each local model.

Response: In Supplementary Note C, we described how we apply the Sobel's test and the joint significance test in each local mediation model.

Please be aware that for some readers, it is very difficult to distinguish the red and purple circles in Supplementary Figure 6. You may wish to consider using shapes as well as colors.

Response: We have changed the purple circle to purple star in this figure (Supplementary Fig. 9 in the revision).

Reviewer #2:

The authors proposed a phylogeny-based method, PhyloMed, to identify human microbes which act as mediators between treatments (or exposures) and health outcomes. PhyloMed uses local mediation models on the internal nodes of the phylogenetic tree. The manuscript is well written. The proposed method outperforms existing methods in simulation studies. I have a couple concerns and hope the authors can clarify and elaborate further.

We thank the reviewer for the encouraging words and constructive input.

1) *The authors claimed that the local models are independent of each other. However, there is a hierarchical order of the phylogenetic tree, the internal nodes are children nodes of other nodes. How could the tests be independent of each other?*

Response: The PhyloMed framework focuses on the hypothesis testing but not estimation (please see our response to comment #3 for the reason). Therefore, we consider what happen under the null (with no mediation) to ensure the validity of the test.

The local mediation tests are asymptotically independent at the internal nodes with no mediation effect (referred to as null internal nodes). This nice property can be achieved because (1) we use the maximum statistic P_{max_j} (between the treatment-mediator association test p-value P_{α_j} and mediator-outcome association test p-value P_{β_j}) as mediation test statistic in each local model and (2) the mediator of the local model at an internal node is the subcomposition (e.g. $(M_j, 1 - M_j)$ in Figure 1) consist of the abundance aggregated at the left and right child nodes (as opposed to the aggregated abundance at the internal node) and the subcompositions over the internal nodes can be modeled as independent variables (please see the justification of subcomposition independence in the first paragraph on page 16).

The p-value under the null is stochastically larger than the p-value under the alternative. Given (1), at the null internal node j (at least one of the α_j and β_j is zero), the P_{max_j} converges to the p-value (P_{α_j} or P_{β_j}) under the null ($\alpha_j = 0$ or $\beta_j = 0$) as the sample size goes to infinity. Given (2), the P_{α_j} 's over internal nodes under $\alpha_j = 0$ are asymptotically independent, and similarly, P_{β_j} 's across internal nodes under $\beta_j = 0$ are asymptotically independent. Hence, P_{max_j} 's over null internal nodes are also asymptotically independent.

For clarity, we have re-written the proof of independence. Please see the Methods section “Mediating subcomposition detection” on page 22.

2) *For the simulation studies, the number of mediating microbes is very small. Is this close to real world? When the number of mediating microbes is large, will FDR be still controlled?*

Response: We have evaluated the performance of FDR control under different density levels of mediation signals (Supplementary Fig. 4). In particular, we considered the setting where 15 out of 100 OTUs have the mediation effect. In this setting, on average, 36% of the 99 internal nodes are mediating nodes since all the ancestor nodes of the mediating OTUs have mediation effects in our simulation. The empirical FDR is controlled for all signal density levels, even though BH controls

the FDR at a more stringent level for denser signals. The additional numerical evaluation has been added at the end of page 8.

3) *The authors did the test for each internal node separately. How will you evaluate the joint direct and indirect mediation effects?*

Response: Evaluating the total mediation effect and direct effect usually requires jointly modeling all the mediators. The PhyloMed framework concerns the mediation effect of the subcompositions on the internal nodes of the phylogenetic tree (as opposed to all the taxa at one taxonomy level). By assuming the subcomposition independence, it is possible to estimate the joint mediation effects by fitting separate mediation models of individual subcomposition under certain causal structure assumptions [1]. However, the true mediation effect (of the unknown absolute abundance) is not identifiable using relative abundance microbiome data and the mediation effect mentioned in the microbiome literature is defined in the context of each method and is different between different methods. For example, the mediation effect in the CMM [2] is defined under the additive log-ratio transformation to the full composition at a taxonomic level, the mediation effect in microHIMA [3,4] is defined under the isometric log-ratio transformation to the full composition at a taxonomic level, and the mediation effect in PhyloMed is defined under the log-ratio transformation to the subcomposition on the internal node of the phylogenetic tree. These different transformations by different methods render the mediation effect estimates among these methods have different interpretations. Given this, our method (and many other existing methods) focus on hypothesis testing rather than estimation. We have expanded our description of different existing methods in the Background section and the reason to focus on hypothesis testing in the last paragraph of the Discussion section (pages 14-15).

[1] Huang YT, Pan WC. Hypothesis test of mediation effect in causal mediation model with high-dimensional continuous mediators. *Biometrics* 2016;72(2): 402-413.

[2] Sohn MB, Li H. Compositional mediation analysis for microbiome studies. *The Annals of Applied Statistics*. 2019;13(1):661-681.

[3] Zhang H, Chen J, Feng Y, et al. Mediation effect selection in high- dimensional and compositional microbiome data. *Statistics in Medicine*. 2021;40(4):885-896.

[4] Zhang H, Chen J, Li Z, et al. Testing for mediation effect with application to human microbiome data. *Statistics in Biosciences*. 2021;13(2):313-328.

4) *For application purpose, using the proposed method, under what situations, leaf nodes are not identified as mediators but the internal nodes are mediators?*

Response: We would like to first clarify that PhyloMed tests the mediation of the subcomposition on internal nodes of the tree and does not test the mediation of the taxa on the leaf nodes. The different columns in the graph below (Supplementary Fig. 6 in the initial submission or Supplementary Fig. 9 in the revision) represent different situations of the ground truth of the presence/absence of mediation at the leaf nodes and at the upper-level nodes. It does not represent different situations of the PhyloMed test results since PhyloMed does not test the mediation effect at the leaf level.

Supplementary Figure 9: Different scenarios of the ground truth of the presence/absence of mediation effects at the leaf-level taxa and the aggregated taxa at the common ancestors. White circles represent leaf-level taxa not associated with the treatment and outcome. Blue circles represent leaf-level taxa only associated with the treatment. Red circles represent leaf-level taxa only associated with the outcome. Purple star represents mediating leaf-level taxon associated with both the treatment and the outcome.

The last column shows that the mediation effect at the ancestor taxon can be formed by aggregating lower-level taxa associated with only the treatment and lower-level taxa associated with only the outcome. This is conceptually different from the false positives as the mediation effect indeed exists at the ancestor node and it is not possible to see the separation of the treatment-mediator association and mediator-outcome association at the leaf-level taxa by only testing the ancestor node (again, PhyloMed does not test mediation at the leaf-level taxa). Hence, the assumption is used to extend the interpretation of the significant mediation signal (discovered in PhyloMed or any other methods) from upper-level taxa to lower-level taxa, but it is not required to establish the validity of the method. We have shown in the real data application a heuristic approach to investigate the mediation effect at few leaf-level taxa under the PhyloMed-identified mediating internal node. This approach is not a rigorous testing procedure (e.g., no guarantee of FDR control), but it provides us some clue about what happened down to the few leaf-level taxa under the PhyloMed-identified mediating internal node.

This issue of “separation of the treatment-mediator association and mediator-outcome association” commonly occurs when interpreting results from the mediation analysis of microbiome data, even when analyzing taxa at one taxonomy level. In the revision, we extensively expanded the discussion of this issue. Please see the details in the last two paragraphs of the Discussion section on pages 14-15.

Reviewer #3:

The authors proposed a phylogeny-based mediation analysis method PhyloMed to account for the compositionality and high-dimensionality of microbiome data, as well as the composite nature of null hypotheses.

We thank the reviewer for the time and constructive input.

Major comments:

1. To test the mediation effect, the authors assumed that the microbiome data follows assumptions of no unmeasured confounders. However, real microbiome data is really complicated and these assumptions might be violated, how to deal with the unmeasured confounders?

Response: We agree that the issue of unmeasured confounding need to be studied and discussed. In our revision, we have discussed this problem and conducted sensitivity analysis for the findings in the two real data applications. The text below was added in the paragraph on pages 12-13:

“The assumption of no unmeasured confounding is critical in obtaining an unbiased estimate of the mediation effect and establishing the causal interpretation in mediation analysis. The correlation ρ between the residuals of the mediator regression model and the outcome regression model is often used to quantify the magnitude of confounding bias on the mediation effect estimate with $\rho = 0$ implying no confounding bias. Therefore, a common way to examine the sensitivity of finding to the violation of the assumption is to evaluate how the mediation effect estimate changes when $|\rho|$ deviates from zero. Supplementary Fig. 8 displays the sensitivity analysis results for the identified mediating nodes highlighted in Fig. 4a and Supplementary Fig. 7 of the two real data analyses. The 90% confidence interval of the estimated mediation effect covers zero when $|\rho| > 0.13$ in the mouse cecal study and $|\rho| > 0.27$ in the human gut study, which means the sign and significance of the estimated mediation effect remain unchanged if $|\rho|$ is not beyond those values. In our analyses, the sample residual correlations at the identified nodes are very close to zero with the absolute value smaller than 10^{-16} , suggesting that we probably do not have a strong confounding bias in our analyses.”

2. As the authors mentioned in the manuscript, Sobel's test and joint significance test tend to produce conservative type I error rate. In order to deal with such issue, several method have been proposed[1-2]. In particular, Liu et al [2] develops DACT to control Type I error rate by estimating the proportions of the three null cases. The authors should claim the differences between PhyloMed and DACT and compare it with DACT in the comparative study.

Response: We thank the reviewer for calling out the recent paper that developed the DACT method. We have cited the paper and summarized the major differences among mediation methods that account for the composite null hypothesis (please see the first paragraph on page 19). We have studied DACT and the following is our observation. First, DACT is not designed for compositional microbiome data. The association from the univariate analysis (testing the relative abundance of one taxon at a time) is biased due to the unit-sum constraint on the proportions of different taxa. That's why PhyloMed applies the log-ratio transformation to the subcomposition on the internal node, and many other mediation methods designed for microbiome data also apply various treatments to the compositionality (e.g., additive log-ratio, isometric log-ratio transformations). It is not clear how to include DCAT in the comparison; more specifically, what transformation of compositional data should be applied to accompany DCAT. We may use DACT to get local

mediation p-value in the PhyloMed framework by feeding the DACT function (<https://github.com/zhonghualiu/DACT>) with P_{α_j} and P_{β_j} generated from the PhyloMed local mediation models. We tested it out in the simulation and the resulting mediation p-values have severely inflated type I error under the null hypothesis H_{01} ($\alpha = 0, \beta \neq 0$) or H_{10} ($\alpha \neq 0, \beta = 0$) (QQ-plots of the mediation p-values under different mediation null hypotheses are shown below). We are not sure about what caused this strange pattern of inflation from reading the DACT paper. The simulation studies in the DACT paper are very large scale (number of mediators=300,000 and sample size = 500-2000) which is realistic for genome-wide epigenetics studies (the application of the DACT) but not so for microbiome studies. Therefore, we suspect that DCAT is not suitable for microbiome mediation analysis. Given these considerations, we currently show these results only in our response, although we are happy to include them in the manuscript's supplement if it would be helpful.

3. The proposed method PhyloMed is a kind of community level test (global test) by combining the testing results on each internal node. Could the authors provide the testing results on the taxon level using the p values on the internal node? If possible, the authors should provide some simulation studies to compare PhyloMed with other methods[3-6].

Response: PhyloMed tests mediation of subcompositions on internal nodes of the tree and does not test the mediation on the leaf-level taxa. In a real data application, we showed a heuristic approach to investigate mediation effect at the leaf-node taxa under the identified internal node. This approach is not a rigorous testing procedure (e.g. not guarantee FDR control) but it provides us some clue about what happen down to the leaf-level under the mediating internal node identified by PhyloMed. Therefore, we focused on the global tests in our simulation studies comparing these methods. In particular, [3,4] provide global tests but [5,6] do not. We compared the global test of [3] (CMM method) in our simulation. CMM can only handle continuous outcome and it often fails to converge when the sample size is smaller than the number of taxa. Therefore, the CMM result is only reported in the setting of continuous outcome and sample size $n=200$ (Figs 2-3). We also tried [4] in our simulation but for most replicates of simulation, it either gives errors or runs more than a day without convergence. Therefore, it is not feasible to compare method [4] in the simulation.

We have applied all these methods [3-6] to the two real datasets. Methods [3,4] fail to converge and methods [5,6] (microHIMA) cannot detect any mediating OTUs (red text on page 9 and page 11). We have described and cited all these different methods in the Background section (page 3-4).

4. The authors developed *PhyloMed* to deal with the compositional and high-dimensional nature of microbiome incorporating phylogenetic tree, while addressing the composite null hypotheses. What if the phylogenetic tree was not provided or misspecified, could *PhyloMed* still control the type I error rate?

Response: The tree structure is only used in the definition of the subcomposition (i.e., how to hierarchically aggregate the leaf-level taxa). Once the subcomposition mediators are defined, the tree structure is not involved in the testing procedure (i.e., generating local mediation p-value, testing global mediation or detecting mediating subcomposition). Therefore, tree misspecification will not affect the validity of the test. However, the power of the test could be affected if the tree is misspecified. For instance, if the mediating taxa are clustered on the true tree but become more scattered on the misspecified tree, signals on the internal nodes may become less condensed and more challenging to detect. Our simulation study shows that *PhyloMed* is still more powerful than the competing methods when the mediating taxa are randomly scattered on the tree (Supplementary Fig. 3). Generally, the phylogenetic tree can be learned from sequence data using hierarchical agglomerative clustering or other more advanced methods. But if a phylogenetic tree is not readily available, *PhyloMed* can be applied to the taxonomy tree, which is almost always provided in microbiome data. Our R package has incorporated this option. We have discussed these points in the second paragraph on page 13.

[1] James Y. Dai, Janet L. Stanford & Michael LeBlanc. A Multiple-Testing Procedure for High-Dimensional Mediation Hypotheses, *Journal of the American Statistical Association*. 2022;117(537):198-213.

[2] Zhonghua Liu, Jincheng Shen, Richard Barfield, Joel Schwartz, Andrea A. Baccarelli & Xihong Lin. Large-Scale Hypothesis Testing for Causal Mediation Effects with Applications in Genome-wide Epigenetic Studies, *Journal of the American Statistical Association*. 2022;117(537):67-81.

[3] SohnMB, LiH. Compositionalmediationanalysisformicrobiomestudies. *The Annals of Applied Statistics*. 2019;13(1):661-681.

[4] Wang C, Hu J, Blaser MJ, et al. Estimating and testing the microbial causal mediation effect with high-dimensional and compositional micro- biome data. *Bioinformatics*. 2020;36(2):347-355.

[5] Zhang H, Chen J, Feng Y, et al. Mediation effect selection in high- dimensional and compositional microbiome data. *Statistics in Medicine*. 2021;40(4):885-896.

[6] Zhang H, Chen J, Li Z, et al. Testing for mediation effect with application to human microbiome data. *Statistics in Biosciences*. 2021;13(2):313-328.

Broad utility and novelty of PhyloMed

We develop PhyloMed to combat low statistical power in the microbiome mediation analysis, especially when the mediation signal is sparse and weak. PhyloMed framework builds upon a phylogeny-guided divide-and-conquer strategy to search for the mediation signals in high-dimensional composition. A new testing procedure is proposed to solve the conservativeness problem in testing composite null hypothesis. These features of PhyloMed are fundamentally different from existing microbiome mediation methods and substantially boost the power of microbiome mediation tests. As a general methodology, PhyloMed can be applied to the mediation analysis of other high-dimensional compositional data. We provide an efficient R package for broad utility of the method.

We thank the reviewers and the Editor for their time and constructive comments. We believe that our revisions and clarifications address the points raised and have substantially improved the manuscript. We hope that you will find the manuscript appropriate for publication in *Genome Biology*.

Sincerely,

Zheng-Zheng Tang

Second round of review

Reviewer 1

The assumption that “there is no complete separation of the two elements” at the lowest level of the tree is too strong and cannot be validated. In many cases, people went a long way to do shotgun metagenomic sequencing in order to profile the microbiome at the species level. It would be a big disappointment to tell them that their data can only be analyzed at the genus level. This is a major limitation of the proposed method that should be acknowledged at the very beginning (e.g., abstract).

Reviewer 2

The authors have addressed my concerns.

Reviewer 3

The authors have addressed my concerns.

Authors response

Response to the comment from Reviewer#1 1. In the Abstract and Background sections, we have clearly stated the PhyloMed’s level of analysis. In the Abstract section, we added “Unlike existing methods that directly identify individual mediating taxa, PhyloMed discovers mediation signals by analyzing subcompositions defined on the phylogenic tree.”. In the Background section (beginning of the 7th paragraph), we added “PhyloMed models mediation effects in many subcompositions on the phylogenetic tree rather than in a full composition with many taxa at the low taxonomic rank.” We have also moved the discussion of the limitation from the Discussion section (the last two paragraphs of the Discussion) to the Background section.